# Viewing the US presidential electoral map through the lens of public health

**Tymor Hamamsy**[1,2]*, **Michael Danziger**[3], **Jonathan Nagler**[1,4], **Richard Bonneau**[1,2,5,6]

**1** Center for Social Media and Politics, NYU, New York, NY, United States of America, **2** Center for Data Science, New York University, New York, NY, United States of America, **3** SUNY Downstate Health Sciences University College of Medicine, Brooklyn, NY, United States of America, **4** Department of Politics, NYU, New York, NY, United States of America, **5** Center for Computational Biology, Flatiron Institute, Simons Foundation, New York, NY, United States of America, **6** Department of Biology, New York University, New York, NY, United States of America

* tymorhamamsy@gmail.com

## Abstract

Health, disease, and mortality vary greatly at the county level, and there are strong geographical trends of disease in the United States. Healthcare is and has been a top priority for voters in the U.S., and an important political issue. Consequently, it is important to determine what relationship voting patterns have with health, disease, and mortality, as doing so may help guide appropriate policy. We performed a comprehensive analysis of the relationship between voting patterns and over 150 different public health and wellbeing variables at the county level, comparing all states, including counties in 2016 battleground states, and counties in states that flipped from majority Democrat to majority Republican from 2012 to 2016. We also investigated county-level health trends over the last 30+ years and find statistically significant relationships between a number of health measures and the voting patterns of counties in presidential elections. Collectively, these data exhibit a strong pattern: counties that voted Republican in the 2016 election had overall worse health outcomes than those that voted Democrat. We hope that this strong relationship can guide improvements in healthcare policy legislation at the county level.

## Introduction

Healthcare is one of the top priorities for voters in the United States [1, 2]. In some 2020 polls, a substantial percentage of Democratic voters indicated it was their top priority [3]. This party-specific preference was reflected in the Democratic primary candidates debates, in which candidates devoted more time to healthcare than any other topic [4]. While still important to Republican voters, it fell behind the issues of terrorism, the economy, social security, immigration, and the military, with an 18 point gap between Democrat and Republican voters on the issue of health care costs and a 13 point gap on the issue of Medicare reform [2]. Prior to the 2016 election, surveys revealed how polarized voters were on health policy issues like the Affordable Care Act (ACA) [5].

In the present study, counties are categorized as "Republican" or "Democrat", referring to the political party that won the majority of votes in the county in the presidential election for

**Data Availability Statement:** The data underlying this study are available on Zenodo (DOI: 10.5281/zenodo.3936108).

**Funding:** RB and TH acknowledge support from the following sources: NIH R01DK103358, Simons

Foundation, NSF- IOS-1546218, R35GM122515, NSF CBET- 1728858, NIH R01AI130945. The funders had no role in study design, data collection and analysis, decision to publish, or preparation of the manuscript.

**Competing interests:** R.B. has ongoing or recent consulting or advisory relationships with Eli Lily, Merus, Merck and Epistemic AI. R.B. has an active research collaboration with Facebook. TH cofounded Fermat's Library. This does not alter our adherence to PLOS ONE policies on sharing data and materials.

that year (i.e. 2012 or 2016). Likewise, when states are referred to as being "Republican" or "Democrat", unless otherwise stated, it is the political party that won the state in the presidential election for that year that is being referenced. "Battleground states" are the tightly contested states of the 2016 Presidential election: Nevada, Colorado, Virginia, Florida, Michigan, Minnesota, Wisconsin, Iowa, New Hampshire, Ohio, and Pennsylvania.

An important prerequisite to understanding the mechanisms driving this significant difference in topical focus is the systematic examination of demographic differences between Republican and Democratic voters [6]. Recent work has demonstrated a contrasting economic realities between Republican and Democratic districts, showing divergence between their economic fortunes and wealth trajectories from 2008 to 2018 [7]. For example, the median household income in districts that voted majority Republican in the midterm elections declined from $55,000 to $53,000, while that of districts that voted majority Democrat experienced an increase from $54,000 to $61,000. Given these divergent economic realities for Republican and Democrat districts, it is not surprising that a recent analysis found that Democratic counties in the 2020 presidential election represented 70% of US GDP [8].

Just as the economy varies greatly by county, so too do health, disease, and mortality. The relationship between voting and health is broad, and previous studies have touched on the effects of health on voter participation, the relationship between life expectancy and voting patterns, and the relationship between health behaviors and voting patterns. Regarding health and voter participation, a study of 30 European countries found that health does have an effect on turnout and that this effect is largest among the elderly [9]. Similarly, a review of 17 studies examining the relationship between voting and health across the US and Europe demonstrated lower voter participation was consistently related to poor self-rated health [10]. Given its relationship to turnout, many analysts have stressed the importance of studying the correlation between health and partisanship in political science research [11]. Many studies, going back several US elections, have also investigated the relationship between health behaviors and voting. Health behavior research is essential because of its fundamental relationship to public health and mortality. For example, a landmark study from 2009 found that smoking and high blood pressure, both of which are preventable, were responsible for the largest number of deaths in the US, and a number of other dietary/lifestyle factors for chronic disease contributed significantly to the number of deaths in the US [12].

The importance of health behaviors to overall health and mortality have made them a popular topic to study alongside voting patterns. One study examining the association between health behavior and the Republican vote share in the 2008 and 2012 Presidential elections found that the Republican vote share was associated with higher odds of flu vaccination and cigarette smoking, but lower odds of avoiding fat/calories, fast/processed foods, and eating fruits and vegetables [13]. Other research found that liberal state ideology was related to lower adult smoking rates, and that this relationship could not be entirely explained by different state anti-smoking policies of the more liberal states [14]. A 2014 study demonstrated that at the state level, there are associations between voting patterns and adolescent vaccination for human papillomavirus (HPV), tetanus-containing (Tdap), and meningococcal (MCV4) vaccinations [15].

While state-level studies are useful, and while there are geographical units in the U.S. smaller than counties (e.g., zip codes), much of the public-health data collection in the U.S., and therefore research, is done at the county-level. And since there are voting data at the county level, an analysis of partisanship and public health at the county level allows for a more granular and nuanced investigation than an analysis at the district or state level. Furthermore, counties are a more natural geographical unit of analysis. As a result, much of the research

investigating the relationship between public health and voting has been done at the county level, and we chose to do so as well.

In the 2012 election, it was found that higher county-level obesity prevalence rates were associated with higher support for the Republican Party Presidential candidate [16]. Earlier research also sought to quantify the extent to which county community health was associated with voting changes between the presidential elections of 2012 and 2016. This earlier research focused on the following variables: physically unhealthy days, mentally unhealthy days, percent food insecure, the teen birth rate, the primary care physician visit rate, the age-adjusted mortality rate, the violent crime rate, average health care costs, the percent diabetic, and the percent overweight or obese in a county [17]. Another county-level study in 2016 looked at Medicare claims data and found that counties with high levels of chronic opioid use were more likely to have voted Republican in the 2016 election, but also that much of that association could be explained by socioeconomic county-level factors [18], while another study used an aggregate measure of well-being from Gallup surveys to relate wellbeing to county level voting, and found that in 2016, counties that shifted from Democrat to Republican had lower wellbeing than those that did not [19].

The association between life expectancy, mortality, and partisanship in the US has previously been studied. Early studies found that the vote for Reagan (Republican) in 1980 was associated with lower mortality in states, and concluded that voting conservative was associated with lower mortality [20]. A later study followed 32,830 participants over a number of years, and found that conservatives and moderates had a greater risk of mortality than liberals, suggesting that party affiliation/political ideology was an associated predictor of mortality [21]. Recently, studies examining county-level trends in mortality rates for the different major causes of death showed strong geographical trends (i.e. regional, spatial patterns); one study found that geographic patterns varied meaningfully by cause of death, and there were clear geographic regions with elevated mortality [22].

Disparities have been demonstrated in life expectancy among U.S. counties over the period from 1980 to 2014 [23]. Part of these geographical differences and county-level inequalities are due to deaths of despair (drug overdose, alcoholic liver disease, and suicide deaths), highlighted as a driving factor in the rising midlife morbidity and mortality among white non-Hispanic Americans [24, 25]. Previous studies have shown strong associations between voting patterns, mortality, health, and disease in the 2016 presidential election. Strong associations have additionally been demonstrated between counties that flipped Republican in 2016 (i.e. those that voted Democrat in 2012) and the rising midlife mortality among white non-Hispanic Americans. This demographic was key to Donald Trump's victory [26]. It has been shown that Trump outperformed Romney (i.e. Trump's vote share for a county in 2016 exceeded Romney's vote share in the 2012 Presidential election) in counties with high drug, alcohol, and suicide mortality rates [27]. A strong association has been shown between life expectancy and both the proportion of votes in a county that went Republican in 2016 as well as the Republican margin shift from 2012 to 2016. This highlights the diverging life expectancies of Republican and Democratic counties and the possible impact of life expectancy on voting behavior [28].

Investigating the relationship between voting patterns at the county level and health, disease, and mortality in the US is important for framing future narratives around healthcare reform. While previous studies have looked at the relationship between voting patterns and life expectancy, mortality risk, and public health variables individually, we performed a comprehensive analysis of the relationship between voting patterns and over 150 different public health and wellbeing variables. Our analysis compares counties in all states, including those in battleground states, and counties in states that flipped from Republican to Democrat from

2012 to 2016, investigating both the relationship of health and wellbeing with the voting margin shifts from 2012 to 2016 as well as overall voting proportions. We believe that investigating associations with shifts in voting and focusing on the battleground and states that flipped can provide discontinuities that allow higher-resolution exploration of associations between political and health outcomes. In addition to comparing recent values of these variables with county-level voting patterns, we examined the dynamics of different public health and wellbeing variables over the last 30+ years. We believe that examining these changes over time can both shed light on a changing electorate and elucidate healthcare trends in counties. Additionally, is our belief that this type of comprehensive exploratory analysis, including broader sets of public health variables than previous studies, can better indicate a clear partisan relationship to the variables examined. relationship between voting patterns in the US and public health, healthcare, life expectancy and mortality rates at the county level. We hope that highlighting these relationships permits better focus of healthcare legislative efforts for counties and that it can inform policy better tailored to the needs of a given locale.

## Materials and methods

In order to show the relationship between voting patterns at the county level and more than 150 different public health, mortality, and life expectancy variables, data from a number of different publicly available sources were aggregated and aligned at the county level. The health and wellbeing data as well as their sources include: diabetes, physical inactivity, and obesity crude rates from the Centers for Disease Control and Prevention diabetes surveillance atlas [29]; mortality rate data from the Global Burden of Disease, including county level mortality rates for a number of respiratory diseases, infectious diseases, cardiovascular diseases, cancers, deaths of despair, and mortality risk at different ages as well as life expectancy [30]; healthcare cost data from the Centers for Medicare and Medicaid Services, including variables such as costs per capita and costs broken down by imaging/drugs/hospice/procedures/dialysis [31]; Medicaid-relevant data collected from the American Community Survey (ACS), including variables about Medicaid usage at the county level [32]; disability-related data also collected from the ACS; Insurance/Uninsurance rate information collected from the Small Area Health Insurance Estimates (SAHIE) [33]; and a number of public health and demographic variables were collected from the County Health Rankings resource, including health behavioral information (i.e. smoking, drinking, and food indices), access to healthcare, and demographic information, among other data [34]. All of these variables were categorized into the following groups: social, physical and economic environment; respiratory diseases; life expectancy and mortality; insurance and healthcare cost; infectious diseases; health outcomes; health behaviors; demographic; deaths of despair; clinical care; cardiovascular diseases; and cancers.

Political voting data at the county level for presidential voting in 2012 and 2016 were collected from the MIT Election project [35]. The margin shift was calculated by taking the difference in the Republican margin (Republican percentage of total vote minus Democratic percentage of total vote) from 2012 to 2016. We define Republican or Democratic counties as those that voted in favor of the Republican or Democratic candidate in 2016. Whenever feasible, data from years as close to 2016 as possible were used (while 2016 data are available for most sources, the GBD data are from 2014).

Pearson correlations and confidence intervals for the correlations between a selection of the collected public health-related variables, the percentage of voters in the county that voted for Donald Trump or Hillary Clinton, and the Republican margin shift were computed from 2012 to 2016 (Table 1). Correlations for counties from all states, counties from battleground states (defined as states that could be reasonably won by either party), and counties from states that

**Table 1. Pearson correlations between different public health-related variables with the percentage of voters in the county that voted for Donald Trump or Hilary Clinton, and the Republican margin shift (from 2012 to 2016).**

| | All States | All States | All States | Battle States | Battle States | Battle States | Flip States | Flip States | Flip States | |
|---|---|---|---|---|---|---|---|---|---|---|
| Variable | % Trump 2016 | % Clinton 2016 | Rep. margin change | % Trump 2016 | % Clinton 2016 | Rep. margin change | % Trump 2016 | % Clinton 2016 | Rep. margin change | category |
| **Asthma** | -0.21 (-0.25 to -0.18) | 0.26 (0.23 to 0.29) | -0.05 (-0.08 to -0.01) | 0.03 (-0.05 to 0.11) | -0.01 (-0.09 to 0.07) | 0 (-0.08 to 0.08) | -0.23 (-0.34 to -0.1) | 0.24 (0.11 to 0.35) | 0.18 (0.06 to 0.3) | Respiratory diseases |
| **Chronic obstructive pulmonary** | 0.46 (0.43 to 0.49) | -0.42 (-0.45 to -0.39) | 0.26 (0.22 to 0.29) | 0.34 (0.26 to 0.4) | -0.3 (-0.37 to -0.23) | 0.12 (0.04 to 0.2) | 0.17 (0.05 to 0.29) | -0.19 (-0.31 to -0.06) | 0.47 (0.37 to 0.57) | Respiratory diseases |
| **Coal workers pneumoconiosis** | 0.1 (0.06 to 0.13) | -0.09 (-0.13 to -0.06) | 0.07 (0.03 to 0.1) | 0.09 (0.01 to 0.17) | -0.06 (-0.14 to 0.02) | 0.12 (0.04 to 0.19) | 0.22 (0.1 to 0.34) | -0.19 (-0.31 to -0.07) | 0.17 (0.04 to 0.29) | Respiratory diseases |
| **Interstitial lung disease** | -0.26 (-0.3 to -0.23) | 0.26 (0.22 to 0.29) | -0.07 (-0.1 to -0.03) | -0.24 (-0.32 to -0.17) | 0.26 (0.19 to 0.34) | -0.1 (-0.18 to -0.03) | -0.3 (-0.41 to -0.18) | 0.27 (0.14 to 0.38) | -0.07 (-0.2 to 0.06) | Respiratory diseases |
| **Mortality risk, age 0–5** | 0.07 (0.04 to 0.11) | 0.02 (-0.02 to 0.05) | 0.01 (-0.03 to 0.05) | -0.08 (-0.16 to 0) | 0.16 (0.08 to 0.24) | -0.17 (-0.25 to -0.09) | -0.1 (-0.22 to 0.03) | 0.15 (0.02 to 0.27) | 0.26 (0.14 to 0.38) | Life expectancy and Mortality |
| **Mortality risk, age 25–45** | 0.11 (0.07 to 0.14) | -0.02 (-0.06 to 0.01) | 0.08 (0.04 to 0.11) | -0.02 (-0.1 to 0.06) | 0.09 (0.01 to 0.17) | -0.05 (-0.13 to 0.03) | 0.06 (-0.07 to 0.18) | 0 (-0.13 to 0.13) | 0.47 (0.37 to 0.57) | Life expectancy and Mortality |
| **Mortality risk, age 45–65** | 0.16 (0.13 to 0.19) | -0.06 (-0.1 to -0.03) | 0.15 (0.11 to 0.18) | 0.01 (-0.07 to 0.08) | 0.08 (0 to 0.15) | 0.01 (-0.07 to 0.09) | 0.07 (-0.06 to 0.19) | -0.02 (-0.14 to 0.11) | 0.47 (0.36 to 0.56) | Life expectancy and Mortality |
| **Mortality risk, age 5–25** | 0.21 (0.18 to 0.25) | -0.13 (-0.17 to -0.1) | 0.11 (0.07 to 0.14) | 0.1 (0.02 to 0.18) | -0.04 (-0.12 to 0.04) | 0.03 (-0.05 to 0.11) | 0.26 (0.14 to 0.38) | -0.22 (-0.34 to -0.1) | 0.58 (0.49 to 0.66) | Life expectancy and Mortality |
| **Mortality risk, age 65–85** | 0.25 (0.21 to 0.28) | -0.17 (-0.2 to -0.13) | 0.2 (0.16 to 0.23) | -0.02 (-0.1 to 0.06) | 0.07 (-0.01 to 0.15) | 0.12 (0.04 to 0.2) | 0.16 (0.03 to 0.28) | -0.13 (-0.26 to -0.01) | 0.43 (0.32 to 0.53) | Life expectancy and Mortality |
| **prct_male_medicaid** | -0.16 (-0.19 to -0.12) | 0.21 (0.18 to 0.24) | 0.19 (0.16 to 0.22) | -0.25 (-0.32 to -0.17) | 0.28 (0.21 to 0.35) | 0.2 (0.13 to 0.28) | 0.03 (-0.09 to 0.16) | -0.03 (-0.16 to 0.1) | 0.53 (0.43 to 0.61) | Insurance and Healthcare cost |
| **prct_female_medicaid** | -0.15 (-0.19 to -0.12) | 0.21 (0.18 to 0.24) | 0.22 (0.18 to 0.25) | -0.23 (-0.31 to -0.16) | 0.27 (0.19 to 0.34) | 0.23 (0.15 to 0.3) | 0.04 (-0.09 to 0.17) | -0.04 (-0.16 to 0.09) | 0.53 (0.43 to 0.61) | Insurance and Healthcare cost |
| **Uninsured %: < = 138% of Poverty** | 0.25 (0.22 to 0.28) | -0.2 (-0.24 to -0.17) | -0.37 (-0.4 to -0.34) | 0.17 (0.09 to 0.25) | -0.1 (-0.18 to -0.02) | -0.44 (-0.5 to -0.37) | -0.13 (-0.26 to -0.01) | 0.11 (-0.02 to 0.23) | -0.12 (-0.24 to 0.01) | Insurance and Healthcare cost |
| **Uninsured %: < = 400% of Poverty** | 0.19 (0.15 to 0.22) | -0.13 (-0.17 to -0.1) | -0.34 (-0.37 to -0.31) | 0.07 (-0.01 to 0.15) | 0 (-0.08 to 0.08) | -0.41 (-0.47 to -0.34) | -0.13 (-0.26 to -0.01) | 0.12 (-0.01 to 0.24) | 0.04 (-0.09 to 0.16) | Insurance and Healthcare cost |
| **Uninsured %: All Incomes** | 0.22 (0.18 to 0.25) | -0.15 (-0.19 to -0.12) | -0.24 (-0.27 to -0.2) | 0.12 (0.04 to 0.2) | -0.04 (-0.12 to 0.04) | -0.29 (-0.36 to -0.22) | 0.02 (-0.11 to 0.14) | -0.03 (-0.16 to 0.1) | 0.31 (0.19 to 0.42) | Insurance and Healthcare cost |
| **Part B Drugs Actual Costs** | -0.33 (-0.36 to -0.29) | 0.33 (0.3 to 0.36) | -0.29 (-0.32 to -0.26) | -0.29 (-0.36 to -0.21) | 0.32 (0.25 to 0.39) | -0.37 (-0.44 to -0.3) | -0.44 (-0.54 to -0.33) | 0.48 (0.38 to 0.57) | -0.47 (-0.56 to -0.36) | Insurance and Healthcare cost |
| **Emergency Department Visits** | -0.39 (-0.42 to -0.36) | 0.4 (0.37 to 0.43) | -0.29 (-0.32 to -0.25) | -0.41 (-0.47 to -0.34) | 0.44 (0.38 to 0.5) | -0.39 (-0.46 to -0.32) | -0.49 (-0.58 to -0.38) | 0.52 (0.42 to 0.61) | -0.39 (-0.49 to -0.27) | Insurance and Healthcare cost |
| **Actual Per Capita Costs** | -0.07 (-0.11 to -0.04) | 0.14 (0.1 to 0.17) | -0.12 (-0.15 to -0.08) | 0 (-0.08 to 0.08) | 0.05 (-0.03 to 0.12) | -0.1 (-0.18 to -0.02) | -0.1 (-0.23 to 0.02) | 0.16 (0.04 to 0.28) | -0.16 (-0.28 to -0.03) | Insurance and Healthcare cost |

*(Continued)*

Table 1. (Continued)

| Variable | All States | All States | All States | Battle States | Battle States | Battle States | Flip States | Flip States | Flip States | |
|---|---|---|---|---|---|---|---|---|---|---|
| | % Trump 2016 | % Clinton 2016 | Rep. margin change | % Trump 2016 | % Clinton 2016 | Rep. margin change | % Trump 2016 | % Clinton 2016 | Rep. margin change | category |
| Percent Male | 0.18 (0.14 to 0.21) | -0.22 (-0.25 to -0.18) | 0.2 (0.16 to 0.23) | 0.13 (0.05 to 0.2) | -0.16 (-0.24 to -0.08) | 0.39 (0.32 to 0.46) | 0.32 (0.2 to 0.43) | -0.33 (-0.44 to -0.21) | 0.52 (0.42 to 0.61) | Insurance and Healthcare cost |
| HIV AIDS | -0.31 (-0.34 to -0.27) | 0.38 (0.35 to 0.41) | -0.21 (-0.24 to -0.17) | -0.1 (-0.18 to -0.03) | 0.16 (0.08 to 0.23) | -0.19 (-0.27 to -0.11) | -0.4 (-0.5 to -0.29) | 0.46 (0.35 to 0.55) | -0.25 (-0.37 to -0.13) | Infectious diseases |
| Lower respiratory infections | 0.13 (0.1 to 0.17) | -0.05 (-0.08 to -0.01) | 0.04 (0 to 0.07) | -0.01 (-0.08 to 0.07) | 0.07 (-0.01 to 0.15) | -0.1 (-0.18 to -0.02) | 0.08 (-0.05 to 0.2) | -0.03 (-0.16 to 0.1) | -0.02 (-0.15 to 0.11) | Infectious diseases |
| Meningitis | -0.17 (-0.21 to -0.14) | 0.27 (0.23 to 0.3) | -0.16 (-0.19 to -0.13) | -0.25 (-0.32 to -0.18) | 0.31 (0.24 to 0.38) | -0.34 (-0.41 to -0.26) | -0.27 (-0.39 to -0.15) | 0.34 (0.23 to 0.45) | -0.07 (-0.2 to 0.05) | Infectious diseases |
| Tuberculosis | -0.36 (-0.39 to -0.33) | 0.44 (0.41 to 0.46) | -0.24 (-0.27 to -0.21) | -0.34 (-0.41 to -0.27) | 0.41 (0.35 to 0.48) | -0.38 (-0.44 to -0.31) | -0.45 (-0.54 to -0.34) | 0.5 (0.4 to 0.59) | -0.19 (-0.31 to -0.06) | Infectious diseases |
| Years of Potential Life Lost Rate | 0.17 (0.13 to 0.2) | -0.09 (-0.13 to -0.06) | 0.21 (0.18 to 0.25) | 0.05 (-0.03 to 0.14) | 0.01 (-0.07 to 0.1) | 0.07 (-0.01 to 0.15) | 0.14 (0.02 to 0.27) | -0.09 (-0.21 to 0.04) | 0.39 (0.28 to 0.49) | Health Outcomes |
| Physically Unhealthy Days | 0.04 (0.01 to 0.08) | 0.02 (-0.01 to 0.06) | 0.12 (0.09 to 0.16) | -0.19 (-0.26 to -0.11) | 0.25 (0.18 to 0.33) | -0.05 (-0.13 to 0.03) | 0.07 (-0.06 to 0.19) | -0.05 (-0.17 to 0.08) | 0.37 (0.25 to 0.47) | Health Outcomes |
| Mentally Unhealthy Days | 0.02 (-0.01 to 0.06) | 0.04 (0 to 0.07) | 0.1 (0.07 to 0.14) | -0.23 (-0.31 to -0.16) | 0.31 (0.24 to 0.38) | -0.11 (-0.19 to -0.03) | 0.12 (-0.01 to 0.24) | -0.08 (-0.21 to 0.05) | 0.25 (0.13 to 0.37) | Health Outcomes |
| Life Expectancy | -0.23 (-0.26 to -0.19) | 0.15 (0.12 to 0.19) | -0.23 (-0.26 to -0.19) | -0.08 (-0.16 to 0) | 0.01 (-0.07 to 0.09) | -0.13 (-0.21 to -0.05) | -0.19 (-0.31 to -0.06) | 0.15 (0.02 to 0.27) | -0.45 (-0.55 to -0.34) | Health Outcomes |
| Life Expectancy (White) | -0.42 (-0.46 to -0.38) | 0.33 (0.29 to 0.38) | -0.38 (-0.42 to -0.34) | -0.37 (-0.46 to -0.27) | 0.28 (0.17 to 0.37) | -0.38 (-0.47 to -0.28) | -0.29 (-0.45 to -0.11) | 0.23 (0.05 to 0.4) | -0.71 (-0.79 to -0.6) | Health Outcomes |
| Age-Adjusted Mortality | 0.2 (0.16 to 0.23) | -0.12 (-0.15 to -0.08) | 0.21 (0.17 to 0.24) | 0.09 (0.01 to 0.16) | -0.02 (-0.1 to 0.06) | 0.08 (0 to 0.16) | 0.17 (0.04 to 0.29) | -0.12 (-0.24 to 0.01) | 0.44 (0.34 to 0.54) | Health Outcomes |
| diabetes_crude | 0.17 (0.14 to 0.2) | -0.1 (-0.13 to -0.06) | 0.23 (0.2 to 0.26) | 0.13 (0.05 to 0.21) | -0.07 (-0.15 to 0.01) | 0.17 (0.09 to 0.25) | 0.39 (0.28 to 0.49) | -0.37 (-0.47 to -0.25) | 0.38 (0.26 to 0.48) | Health Behaviors |
| obesity_crude | 0.16 (0.12 to 0.19) | -0.1 (-0.14 to -0.07) | 0.28 (0.24 to 0.31) | 0.2 (0.12 to 0.27) | -0.17 (-0.25 to -0.09) | 0.33 (0.25 to 0.39) | 0.26 (0.14 to 0.38) | -0.26 (-0.38 to -0.14) | 0.38 (0.27 to 0.49) | Health Behaviors |
| physical_inactivity_crude | 0.36 (0.33 to 0.39) | -0.28 (-0.31 to -0.25) | 0.3 (0.26 to 0.33) | 0.38 (0.31 to 0.45) | -0.31 (-0.38 to -0.24) | 0.2 (0.13 to 0.28) | 0.46 (0.35 to 0.55) | -0.43 (-0.53 to -0.32) | 0.53 (0.44 to 0.62) | Health Behaviors |
| % Smokers | 0.12 (0.09 to 0.16) | -0.04 (-0.07 to 0) | 0.35 (0.32 to 0.38) | -0.03 (-0.11 to 0.05) | 0.09 (0.01 to 0.17) | 0.14 (0.06 to 0.22) | 0.04 (-0.09 to 0.17) | -0.04 (-0.16 to 0.09) | 0.43 (0.32 to 0.53) | Health Behaviors |
| Food Environment Index | 0.06 (0.02 to 0.09) | -0.1 (-0.13 to -0.06) | 0.06 (0.03 to 0.1) | -0.01 (-0.09 to 0.07) | -0.02 (-0.1 to 0.06) | 0.16 (0.08 to 0.24) | 0.17 (0.05 to 0.3) | -0.2 (-0.32 to -0.07) | -0.19 (-0.31 to -0.06) | Health Behaviors |
| % Excessive Drinking | -0.16 (-0.19 to -0.12) | 0.11 (0.07 to 0.14) | 0.07 (0.03 to 0.1) | -0.12 (-0.2 to -0.04) | 0.05 (-0.03 to 0.13) | 0.18 (0.1 to 0.26) | -0.34 (-0.44 to -0.22) | 0.3 (0.18 to 0.41) | -0.25 (-0.36 to -0.12) | Health Behaviors |

(Continued)

**Table 1.** (Continued)

| Variable | All States | All States | All States | Battle States | Battle States | Battle States | Flip States | Flip States | Flip States | |
|---|---|---|---|---|---|---|---|---|---|---|
| | % Trump 2016 | % Clinton 2016 | Rep. margin change | % Trump 2016 | % Clinton 2016 | Rep. margin change | % Trump 2016 | % Clinton 2016 | Rep. margin change | category |
| **% Food Insecure** | -0.14 (-0.17 to -0.1) | 0.21 (0.18 to 0.24) | -0.07 (-0.11 to -0.04) | -0.21 (-0.28 to -0.13) | 0.28 (0.21 to 0.35) | -0.23 (-0.3 to -0.15) | -0.15 (-0.27 to -0.02) | 0.17 (0.05 to 0.29) | 0.2 (0.08 to 0.32) | Health Behaviors |
| **Drug Overdose Mortality Rate** | 0.15 (0.11 to 0.2) | -0.13 (-0.18 to -0.09) | 0.3 (0.25 to 0.34) | 0.12 (0.02 to 0.21) | -0.07 (-0.17 to 0.03) | 0.11 (0.01 to 0.21) | 0.02 (-0.13 to 0.16) | 0.06 (-0.09 to 0.2) | 0.05 (-0.1 to 0.19) | Health Behaviors |
| **% Insufficient Sleep** | -0.16 (-0.19 to -0.12) | 0.25 (0.22 to 0.28) | 0.03 (-0.01 to 0.06) | -0.27 (-0.35 to -0.2) | 0.36 (0.29 to 0.43) | -0.03 (-0.11 to 0.05) | 0.09 (-0.04 to 0.22) | -0.05 (-0.17 to 0.08) | 0.14 (0.02 to 0.26) | Health Behaviors |
| **opioid_prescribing_rate** | 0.13 (0.09 to 0.16) | -0.1 (-0.13 to -0.06) | -0.02 (-0.05 to 0.02) | 0.1 (0.02 to 0.18) | -0.04 (-0.12 to 0.04) | -0.06 (-0.14 to 0.02) | 0.13 (0 to 0.25) | -0.13 (-0.25 to 0) | 0.16 (0.03 to 0.28) | Health Behaviors |
| **Alcohol use disorders** | -0.23 (-0.26 to -0.19) | 0.18 (0.15 to 0.21) | -0.02 (-0.06 to 0.01) | -0.3 (-0.37 to -0.22) | 0.27 (0.2 to 0.34) | -0.02 (-0.1 to 0.06) | -0.4 (-0.5 to -0.28) | 0.4 (0.29 to 0.5) | 0.09 (-0.04 to 0.21) | Deaths of Despair |
| **Drug use disorders** | 0.11 (0.07 to 0.14) | -0.08 (-0.12 to -0.05) | 0.09 (0.06 to 0.12) | -0.15 (-0.23 to -0.07) | 0.2 (0.12 to 0.27) | -0.1 (-0.18 to -0.02) | -0.06 (-0.18 to 0.07) | 0.11 (-0.02 to 0.24) | 0.04 (-0.09 to 0.17) | Deaths of Despair |
| **Interpersonal violence** | -0.29 (-0.32 to -0.26) | 0.37 (0.34 to 0.4) | -0.14 (-0.17 to -0.1) | -0.3 (-0.37 to -0.22) | 0.37 (0.3 to 0.43) | -0.3 (-0.37 to -0.22) | -0.36 (-0.47 to -0.24) | 0.42 (0.31 to 0.52) | -0.09 (-0.21 to 0.04) | Deaths of Despair |
| **% With Access** | -0.37 (-0.4 to -0.34) | 0.32 (0.28 to 0.35) | -0.25 (-0.28 to -0.22) | -0.41 (-0.48 to -0.35) | 0.39 (0.32 to 0.46) | -0.21 (-0.28 to -0.13) | -0.28 (-0.4 to -0.16) | 0.28 (0.16 to 0.4) | -0.38 (-0.48 to -0.26) | Clinical Care |
| **PCP Rate** | -0.35 (-0.38 to -0.32) | 0.31 (0.28 to 0.35) | -0.27 (-0.3 to -0.24) | -0.3 (-0.37 to -0.23) | 0.27 (0.2 to 0.35) | -0.21 (-0.28 to -0.13) | -0.35 (-0.46 to -0.23) | 0.34 (0.22 to 0.45) | -0.38 (-0.48 to -0.26) | Clinical Care |
| **Dentist Rate** | -0.38 (-0.41 to -0.35) | 0.34 (0.31 to 0.37) | -0.23 (-0.27 to -0.2) | -0.35 (-0.42 to -0.28) | 0.32 (0.25 to 0.39) | -0.24 (-0.31 to -0.16) | -0.47 (-0.57 to -0.37) | 0.49 (0.39 to 0.58) | -0.41 (-0.51 to -0.29) | Clinical Care |
| **MHP Rate** | -0.4 (-0.43 to -0.37) | 0.35 (0.32 to 0.39) | -0.23 (-0.26 to -0.19) | -0.47 (-0.53 to -0.41) | 0.46 (0.39 to 0.52) | -0.26 (-0.33 to -0.18) | -0.52 (-0.61 to -0.42) | 0.5 (0.4 to 0.59) | -0.35 (-0.46 to -0.23) | Clinical Care |
| **Preventable Hosp. Rate** | 0.13 (0.09 to 0.16) | -0.05 (-0.09 to -0.02) | 0.14 (0.11 to 0.17) | -0.05 (-0.12 to 0.03) | 0.08 (0 to 0.16) | 0.16 (0.08 to 0.23) | -0.02 (-0.15 to 0.1) | 0.06 (-0.07 to 0.18) | 0.15 (0.03 to 0.27) | Clinical Care |
| **% Screened** | -0.17 (-0.21 to -0.14) | 0.16 (0.12 to 0.19) | 0.09 (0.06 to 0.13) | -0.11 (-0.19 to -0.03) | 0.1 (0.02 to 0.18) | 0.12 (0.04 to 0.2) | -0.01 (-0.14 to 0.12) | -0.03 (-0.15 to 0.1) | -0.1 (-0.23 to 0.03) | Clinical Care |
| **% Vaccinated** | -0.23 (-0.26 to -0.19) | 0.21 (0.18 to 0.25) | -0.1 (-0.13 to -0.06) | -0.35 (-0.42 to -0.28) | 0.35 (0.28 to 0.42) | -0.26 (-0.33 to -0.18) | -0.34 (-0.45 to -0.22) | 0.34 (0.23 to 0.45) | -0.51 (-0.6 to -0.41) | Clinical Care |
| **Cardiomyopathy & myocarditis** | -0.2 (-0.23 to -0.16) | 0.27 (0.24 to 0.31) | -0.05 (-0.08 to -0.01) | -0.25 (-0.32 to -0.17) | 0.34 (0.27 to 0.41) | -0.17 (-0.24 to -0.09) | 0 (-0.13 to 0.13) | 0.04 (-0.09 to 0.16) | -0.09 (-0.21 to 0.04) | Cardiovascular diseases |
| **Cardiovascular diseases** | 0.23 (0.2 to 0.26) | -0.14 (-0.17 to -0.11) | 0.2 (0.17 to 0.24) | 0.06 (-0.01 to 0.14) | 0.01 (-0.07 to 0.09) | 0.18 (0.11 to 0.26) | 0.26 (0.14 to 0.38) | -0.22 (-0.34 to -0.09) | 0.42 (0.3 to 0.52) | Cardiovascular diseases |
| **Hypertensive heart disease** | -0.13 (-0.16 to -0.09) | 0.18 (0.15 to 0.22) | -0.13 (-0.17 to -0.1) | -0.19 (-0.26 to -0.11) | 0.23 (0.15 to 0.3) | -0.22 (-0.3 to -0.14) | -0.31 (-0.42 to -0.19) | 0.32 (0.2 to 0.43) | -0.12 (-0.24 to 0.01) | Cardiovascular diseases |

(*Continued*)

**Table 1.** (Continued)

| Variable | All States<br>% Trump 2016 | All States<br>% Clinton 2016 | All States<br>Rep. margin change | Battle States<br>% Trump 2016 | Battle States<br>% Clinton 2016 | Battle States<br>Rep. margin change | Flip States<br>% Trump 2016 | Flip States<br>% Clinton 2016 | Flip States<br>Rep. margin change | category |
|---|---|---|---|---|---|---|---|---|---|---|
| Ischemic heart disease | 0.28 (0.25 to 0.31) | -0.2 (-0.23 to -0.16) | 0.26 (0.22 to 0.29) | 0.14 (0.06 to 0.21) | -0.06 (-0.14 to 0.01) | 0.24 (0.17 to 0.32) | 0.33 (0.21 to 0.44) | -0.29 (-0.4 to -0.17) | 0.47 (0.36 to 0.56) | Cardiovascular diseases |
| Liver cancer | -0.18 (-0.21 to -0.14) | 0.25 (0.22 to 0.28) | -0.13 (-0.16 to -0.1) | -0.28 (-0.35 to -0.2) | 0.32 (0.25 to 0.39) | -0.12 (-0.2 to -0.04) | -0.26 (-0.38 to -0.14) | 0.32 (0.2 to 0.43) | 0.24 (0.12 to 0.36) | Cancers |
| Malignant skin melanoma | 0.54 (0.51 to 0.56) | -0.56 (-0.59 to -0.54) | 0.15 (0.11 to 0.18) | 0.46 (0.4 to 0.52) | -0.44 (-0.5 to -0.37) | -0.06 (-0.13 to 0.02) | 0.34 (0.22 to 0.45) | -0.36 (-0.46 to -0.24) | 0.14 (0.02 to 0.27) | Cancers |
| Stomach cancer | -0.35 (-0.38 to -0.32) | 0.44 (0.41 to 0.47) | -0.16 (-0.2 to -0.13) | -0.47 (-0.53 to -0.4) | 0.52 (0.46 to 0.57) | -0.13 (-0.21 to -0.05) | -0.3 (-0.41 to -0.18) | 0.37 (0.25 to 0.47) | 0.03 (-0.09 to 0.16) | Cancers |
| Testicular cancer | 0.31 (0.28 to 0.34) | -0.29 (-0.32 to -0.25) | 0.27 (0.24 to 0.3) | 0.25 (0.17 to 0.32) | -0.23 (-0.31 to -0.16) | 0.21 (0.13 to 0.29) | 0.42 (0.31 to 0.52) | -0.41 (-0.51 to -0.3) | 0.52 (0.43 to 0.61) | Cancers |

Correlations for counties from all states, counties from battleground states, and counties from states that flipped from Democratic in 2012 to Republican in 2016, are presented.

"flipped" from Democrat in 2012 to Republican in 2016, are presented. The battleground states are: Nevada, Colorado, Virginia, Florida, Michigan, Minnesota, Wisconsin, Iowa, New Hampshire, Ohio, and Pennsylvania. The flipped states are: Michigan, Pennsylvania, Wisconsin and Maine. In S1 Table, the same data as Table 1 are presented with the inclusion of the remaining public health-related variables collected. In S2 Table, we use the same data and structure as S1 Table, except that the correlations are now weighed correlations, where the weights are equal to the base 10 logarithm of the population of the county.

Every county was assigned as either Republican or Democratic depending on the majority vote in 2016, and the mean, median, 1st quartile, and 3rd quartile values for different public health-related variables were calculated. The differences in these values for Republican and Democratic counties are presented in Table 2, along with the Student t-test statistics and p values for the mean comparisons. S3 Table presents the same data as Table 2, except we additionally include the remaining public health-related variables that we collected.

The distributions of different public health variables for Republican and Democratic counties are presented in Fig 1. The dynamics of different public health variables over time for counties based on their 2016 political party are compared across the aggregated data (Fig 2). The Republican margin shift was then compared with different public health variables for counties in states that flipped from Democratic in 2012 to Republican in 2016, indicating the 2016 total number of votes and the 2016 election outcome for counties by the size and color of their points, respectively (Fig 3).

In the next part of our analysis, multivariate linear models were built to predict the percentage of voters in the county that voted for Donald Trump or Hillary Clinton, and the Republican margin shift. For every public health variable under consideration, we built a linear model to predict the 3 outcomes, using that variable as well as the following education, socio-economic status, and demographic control variables: "Graduation Rate", "% Some College", "% Non-Hispanic White", "% 65 and over", "Household Income", "% Severe Housing Cost

**Table 2. Quantile and mean comparisons of Republican and Democratic counties across select public-health measures.**

| Variable | % Mean Difference | % Median Difference | % Top Quartile Difference | % Bottom Quartile Difference | t-test p value | t-test t statistic | category |
|---|---|---|---|---|---|---|---|
| **Asthma** | -15.53% | -2.09% | -24.20% | 6% | 1.75E-12 | -7.227185 | Respiratory diseases |
| **Interstitial lung disease** | -7.59% | -7.41% | -6.82% | -6% | 6.27E-11 | -6.661729 | Respiratory diseases |
| **Chronic obstructive pulmonary** | 33.27% | 31.82% | 31.19% | 33% | 2.00E-85 | 22.186229 | Respiratory diseases |
| **Coal workers pneumoconiosis** | 360.40% | 50.00% | 75.00% | 0% | 7.99E-10 | 6.1667217 | Respiratory diseases |
| **Mortality risk, age 0–5** | -2.31% | 11.48% | -10.87% | 17% | 0.252621008 | -1.145192 | Life expectancy and Mortality |
| **Mortality risk, age 25–45** | 0.57% | 11.03% | -8.44% | 19% | 0.769660064 | 0.2929605 | Life expectancy and Mortality |
| **Mortality risk, age 45–65** | 3.77% | 9.50% | -1.84% | 13% | 0.012094384 | 2.5172813 | Life expectancy and Mortality |
| **Mortality risk, age 65–85** | 5.05% | 6.41% | 2.53% | 7% | 2.98E-13 | 7.460388 | Life expectancy and Mortality |
| **Mortality risk, age 5–25** | 8.86% | 24.67% | -0.44% | 32% | 4.31E-05 | 4.1224536 | Life expectancy and Mortality |
| **Part B Drugs Actual Costs** | -97.17% | -85.75% | -91.46% | -60% | 0.00270063 | -3.012516 | Insurance and Healthcare cost |
| **Emergency Department Visits** | -97.13% | -81.77% | -89.77% | -65% | 0.002669971 | -3.016033 | Insurance and Healthcare cost |
| **prct_male_medicaid** | -12.54% | -9.83% | -15.64% | -5% | 2.80E-10 | -6.419353 | Insurance and Healthcare cost |
| **prct_female_medicaid** | -11.51% | -7.56% | -15.69% | -5% | 2.46E-09 | -6.05624 | Insurance and Healthcare cost |
| **Actual Per Capita Costs** | -5.00% | -3.64% | -7.12% | -3% | 1.83E-10 | -6.464521 | Insurance and Healthcare cost |
| **Percent Male** | 2.11% | 2.24% | 1.60% | 2% | 4.33E-20 | 9.4117126 | Insurance and Healthcare cost |
| **Uninsured %: < = 400% of Poverty** | 2.67% | -1.50% | 3.01% | -2% | 0.171888273 | 1.367543 | Insurance and Healthcare cost |
| **Uninsured %: All Incomes** | 6.38% | 1.96% | 4.48% | 11% | 0.005906647 | 2.7616682 | Insurance and Healthcare cost |
| **Uninsured %: < = 138% of Poverty** | 9.40% | 11.25% | 12.44% | 0% | 3.19E-06 | 4.6948186 | Insurance and Healthcare cost |
| **HIV AIDS** | -58.75% | -57.86% | -67.85% | -45% | 3.27E-30 | -12.12769 | Infectious diseases |
| **Tuberculosis** | -43.03% | -40.74% | -46.67% | -35% | 1.28E-35 | -13.4303 | Infectious diseases |
| **Meningitis** | -16.06% | -6.98% | -25.40% | 0% | 2.10E-17 | -8.782586 | Infectious diseases |
| **Lower respiratory infections** | 2.62% | 1.57% | -1.82% | 9% | 0.124374871 | 1.5385809 | Infectious diseases |
| **Life Expectancy (White)** | -2.28% | -2.57% | -2.97% | -2% | 1.34E-21 | -9.967263 | Health Outcomes |
| **Life Expectancy** | -1.41% | -1.82% | -2.21% | -1% | 1.07E-08 | -5.804499 | Health Outcomes |
| **Mentally Unhealthy Days** | -0.59% | 0.76% | 1.82% | -3% | 0.399461955 | -0.843066 | Health Outcomes |
| **Physically Unhealthy Days** | -0.59% | 0.99% | -0.48% | 1% | 0.530765566 | -0.627172 | Health Outcomes |
| **Years of Potential Life Lost Rate** | 5.60% | 16.53% | 0.73% | 23% | 0.00682826 | 2.7151188 | Health Outcomes |
| **Age-Adjusted Mortality** | 7.07% | 16.06% | 1.65% | 22% | 1.27E-04 | 3.8574268 | Health Outcomes |
| **% Food Insecure** | -13.59% | -4.74% | -19.69% | 1% | 2.84E-12 | -7.147556 | Health Behaviors |
| **% Insufficient Sleep** | -5.18% | -5.05% | -6.50% | -4% | 4.73E-14 | -7.719342 | Health Behaviors |
| **% Excessive Drinking** | -2.24% | -4.20% | -4.44% | 1% | 0.031192086 | -2.159568 | Health Behaviors |
| **% Smokers** | 4.07% | 5.65% | 1.45% | 10% | 0.001194561 | 3.2560056 | Health Behaviors |
| **Food Environment Index** | 5.89% | 2.67% | -0.30% | 11% | 4.28E-08 | 5.5554343 | Health Behaviors |
| **obesity_crude** | 8.91% | 12.63% | 2.59% | 20% | 2.73E-14 | 7.8114067 | Health Behaviors |

*(Continued)*

**Table 2.** (Continued)

| Variable | % Mean Difference | % Median Difference | % Top Quartile Difference | % Bottom Quartile Difference | t-test p value | t-test t statistic | category |
|---|---|---|---|---|---|---|---|
| **diabetes_crude** | 14.08% | 21.43% | 12.33% | 20% | 9.06E-13 | 7.289131 | Health Behaviors |
| **Drug Overdose Mortality Rate** | 14.97% | 13.37% | 11.53% | 15% | 6.15E-06 | 4.5607718 | Health Behaviors |
| **opioid_prescribing_rate** | 16.94% | 23.90% | 25.76% | 13% | 3.90E-09 | 5.9619381 | Health Behaviors |
| **physical_inactivity_crude** | 16.97% | 18.88% | 11.79% | 29% | 2.80E-32 | 12.557282 | Health Behaviors |
| **% 65 and over** | 19.68% | 21.38% | 17.73% | 25% | 1.62E-47 | 15.595938 | Demographic |
| **% Non-Hispanic White** | 50.20% | 69.16% | 22.38% | 102% | 2.72E-85 | 23.428989 | Demographic |
| **% Rural** | 93.91% | 233.63% | 80.54% | 788% | 2.51E-66 | 19.412684 | Demographic |
| **Interpersonal violence** | -38.67% | -27.68% | -41.59% | -9% | 3.10E-23 | -10.42015 | Deaths of Despair |
| **Alcohol use disorders** | -25.97% | -21.51% | -19.34% | -16% | 4.65E-08 | -5.545754 | Deaths of Despair |
| **Drug use disorders** | 8.55% | 2.25% | 16.33% | -4% | 0.001438594 | 3.1974805 | Deaths of Despair |
| **MHP Rate** | -52.14% | -60.95% | -50.92% | -66% | 7.32E-37 | -13.68433 | Clinical Care |
| **Dentist Rate** | -36.14% | -41.90% | -33.63% | -40% | 2.84E-35 | -13.25697 | Clinical Care |
| **PCP Rate** | -33.90% | -37.34% | -34.07% | -33% | 1.03E-32 | -12.68707 | Clinical Care |
| **% With Access** | -18.96% | -24.06% | -18.62% | -25% | 1.26E-27 | -11.43411 | Clinical Care |
| **% Vaccinated** | -6.02% | -4.55% | -4.08% | -8% | 3.03E-08 | -5.60294 | Clinical Care |
| **% Screened** | -0.95% | -2.44% | 0.00% | -3% | 0.305959851 | -1.024512 | Clinical Care |
| **Preventable Hosp. Rate** | 5.12% | 6.49% | 3.77% | 13% | 0.026039081 | 2.2310975 | Clinical Care |
| **Hypertensive heart disease** | -22.88% | -17.79% | -26.35% | -14% | 2.03E-09 | -6.090275 | Cardiovascular diseases |
| **Cardiomyopathy & myocarditis** | -13.50% | -15.48% | -16.10% | -6% | 2.64E-11 | -6.791812 | Cardiovascular diseases |
| **Cardiovascular diseases** | 7.69% | 11.17% | 4.65% | 15% | 2.99E-09 | 6.0209145 | Cardiovascular diseases |
| **Ischemic heart disease** | 13.42% | 16.28% | 13.25% | 20% | 1.94E-16 | 8.4505235 | Cardiovascular diseases |
| **Stomach cancer** | -20.13% | -18.81% | -25.95% | -13% | 1.79E-42 | -14.92076 | Cancers |
| **Liver cancer** | -12.99% | -12.05% | -12.69% | -10% | 2.68E-20 | -9.572363 | Cancers |
| **Testicular cancer** | 16.58% | 21.74% | 18.52% | 25% | 9.01E-27 | 11.261797 | Cancers |
| **Malignant skin melanoma** | 26.76% | 27.04% | 22.20% | 31% | 9.65E-98 | 24.990984 | Cancers |

Every county was assigned as either Republican or Democratic depending on the majority vote in 2016, and the mean, median, 1st quartile, and 3rd quartile values for different public health-related variables were calculated. The differences in these values for Republican and Democratic counties are presented in Table 2, along with the Student t-test statistics and p values for the mean comparisons.

Burden", "% Rural", "Actual Per Capita Costs", and "Percent Male". Before applying these linear models, we normalized all of the covariates to have a standard deviation of one and a mean of zero. S4 Table reports the coefficients for each public health variable under consideration, as well as the standard error for those coefficients, for each predictive model.

As many of the public health variables and control variables that we collected are correlated with one another, the next part of our analysis involved attempting to tease out the most important variables and categories in the relationship between health and voting. Before studying the importance of different variables, we first grouped them into their natural categories. For each category, using all of the variables in the category, we calculated the principal components. Fig 4 shows a plot of the first 2 principal components for 9 different categories of variables, with every county colored to indicate the 2012 to 2016 presidental outcome. For every category, we next applied lasso regression to predict the percentage of voters in the county that

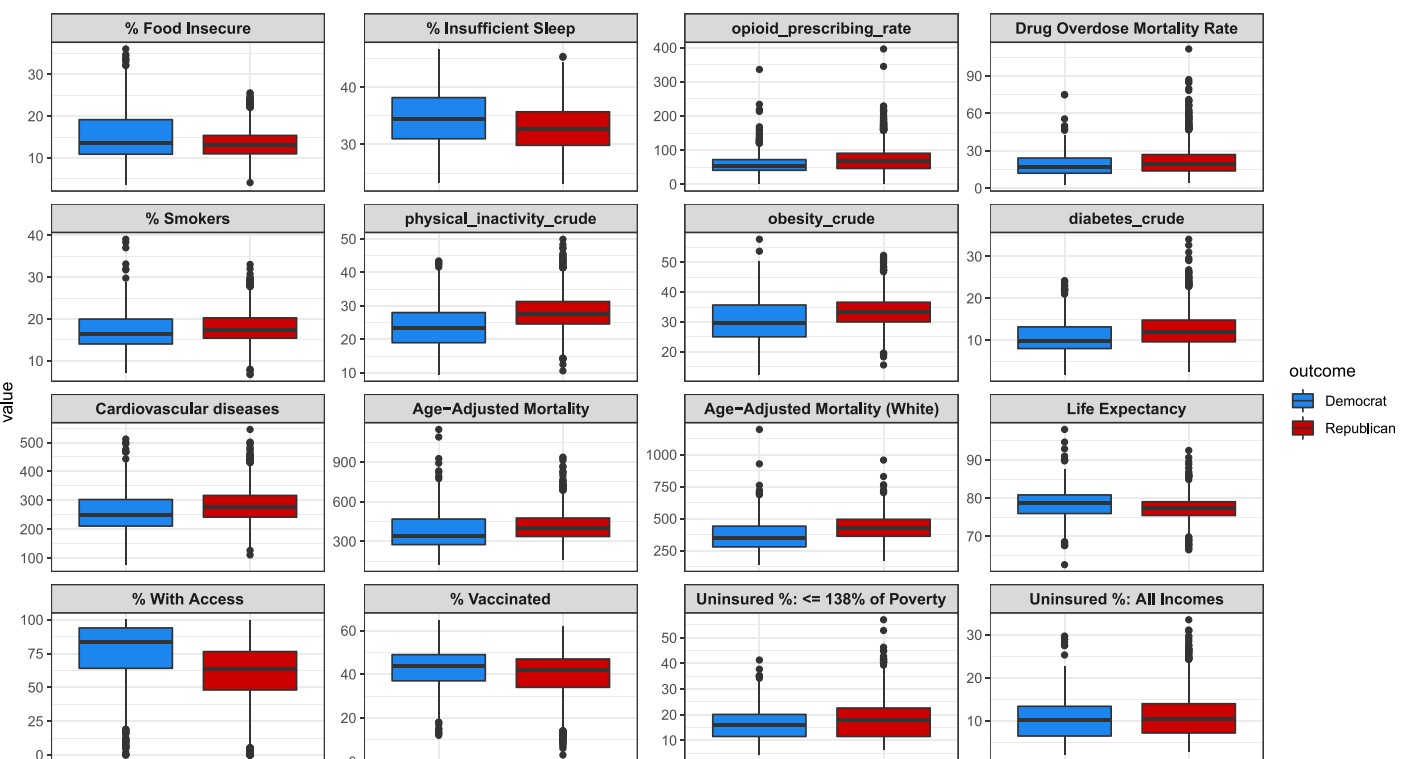

**Fig 1. Boxplots comparing select public health variables for Democratic and Republican counties.** As shown, there are higher rates of lifestyle factors like smoking, obesity, and physical inactivity, and chronic diseases that are affected by lifestyle like cardiovascular diseases and diabetes in Republican counties than in Democratic counties. Democratic counties also have higher life expectancy, insurance rates, and lower mortality rates than Republican counties. The percentage of individuals who are food insecure or get insufficient sleep in Democratic counties is higher than in Republican counties.

voted for Donald Trump or Hillary Clinton, and the Republican margin shift. Lasso regression is a form of linear regression that involves a shrinkage regularization term; this term performs both variable selection as well as regularization, by shrinking some coefficients to 0 [36]. For every category, we used the variables from that category as well as the education, socio-economic status, and demographic control variables as covariates. The variable importance of every variable was calculated, which in lasso regression, is the ranked absolute value of the coefficeints from the final model. Table 3 shows the variable importance of the variables from each category for predicting the percentage of voters in the county that voted for Donald Trump or Hillary Clinton, and the Republican margin shift.

## Data and code availability

The analysis and code from this manuscript can be found at the following link: https://github.com/tymor22/Health-and-Politics/. All of the data analyzed in this manuscript is available at the following link: https://zenodo.org/record/3936108#.Xyc5O_hKh_Q with the DOI number 10.5281/zenodo.3936108. The R programming language was used to conduct all of the data cleaning, modelling, analysis, and plotting.

## Results

Our analysis covers 3,156 counties from all 50 states and Washington DC, of which 2650 went Republican and 506 went Democratic in 2016. These counties exhibit significantly different

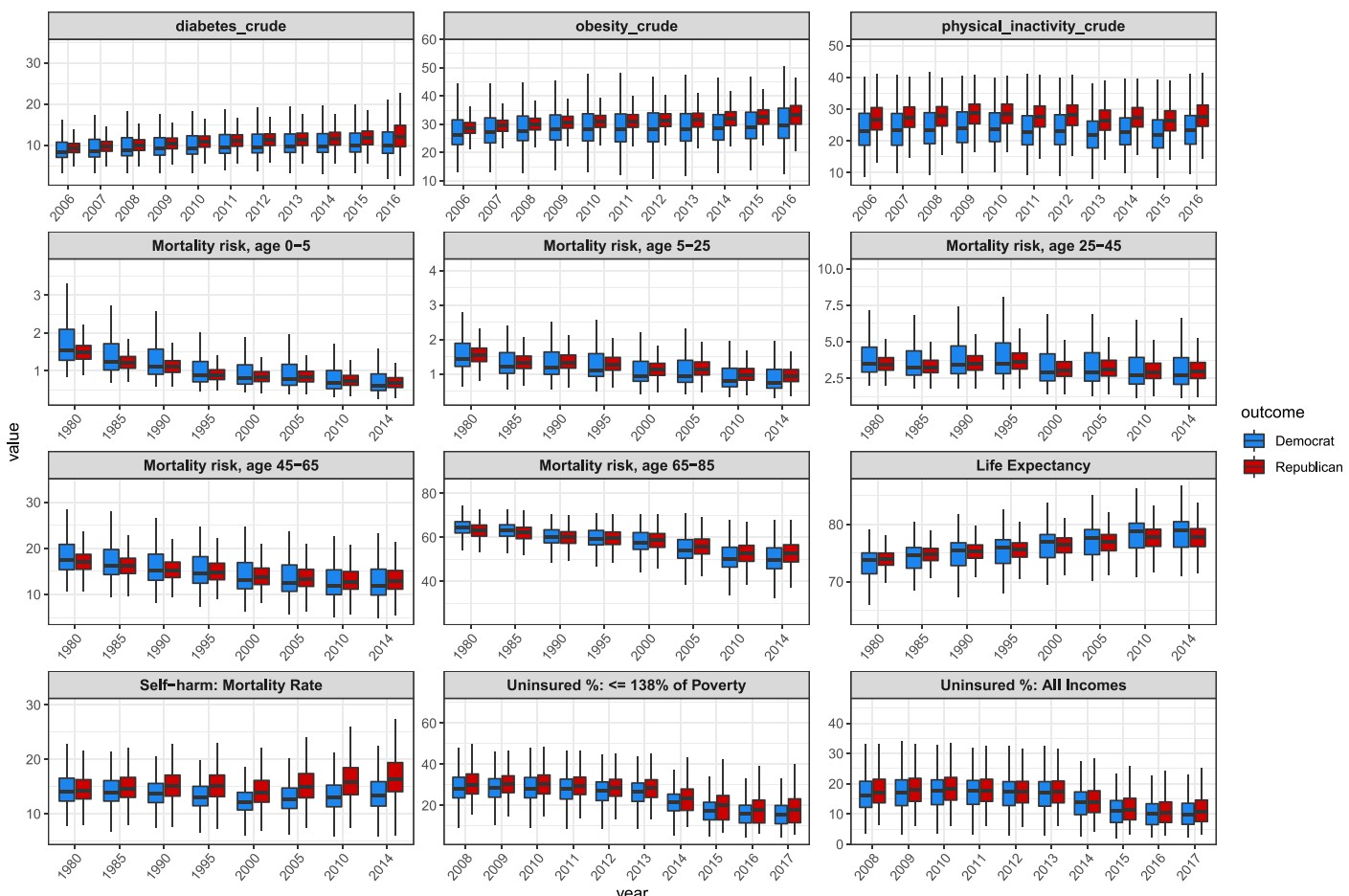

**Fig 2. Boxplots comparing Democratic and Republican counties (defined by 2016 presidential election voting) over time for a number of public health variables.**
For diabetes, obesity, and physical inactivity, there has been a growing divide between Republican and Democratic counties between 2006 and 2016. For mortality risk across every age group, Democratic counties have improved more than Republican counties over the time period from 1980 to 2014.

demographics: the average "% 65 and older" in Republican counties was 19.68% higher compared to Democratic counties; the average "% Non-Hispanic White" in Republican counties was 50.20% higher; the average "% Rural" in Republican counties was 93.91% higher; and the average "% with Some College" in Republican counties was 8.51% lower. These demographic differences are also driving healthcare differences. For example, Republican counties received 52% of Medicare funding (of which patients over 65 account for 85% [37]) in 2017, compared to 50.5% of spending in 2007. Additionally, the total number of non-elderly individuals with preexisting conditions in states that voted Republican in 2016 was 74.3 million, compared to 59.4 million in Democratic states. However, the average percentage of non-elderly people with preexisting conditions in Republican states was 50% compared to 51% in Democratic states.

Table 1 shows Pearson correlations between different public health-related variables and the percentage of voters in counties who voted for Trump in 2016, for Clinton in 2016, and the Republican margin change from 2012 to 2016. These correlations were calculated for all states, for 2016 battleground states, and for states that flipped in 2016. For counties in all states, the percentage of votes for Trump had a correlation with the life expectancy of whites of -.42, and a correlation with physical inactivity of .36. Some of the variables that are highly correlated

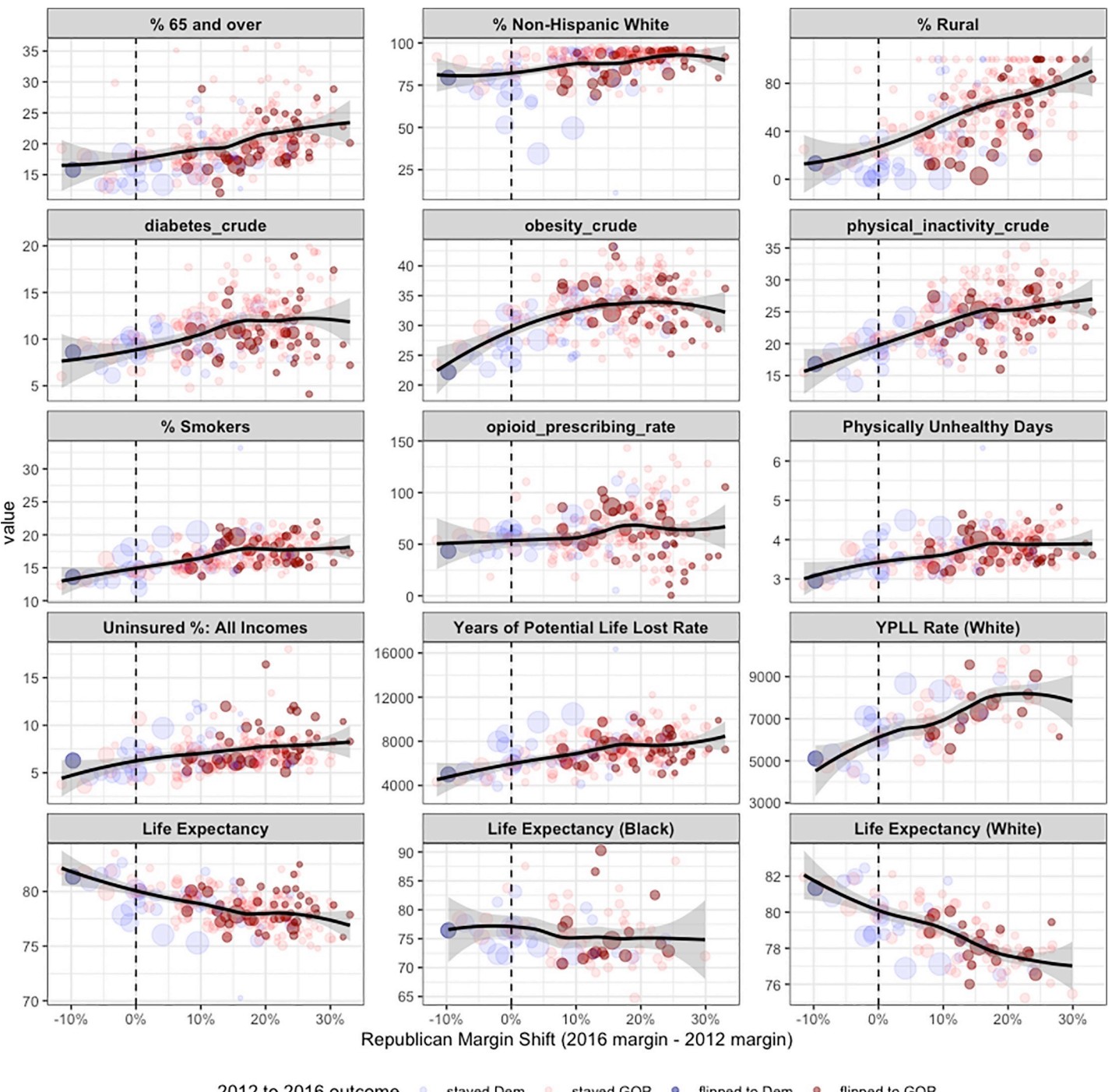

**Fig 3. Scatterplots for all counties in the 4 states that flipped from Democrat to Republican in 2016 (Michigan, Pennsylvania, Wisconsin, Maine), showing the Republican margin shift on the x axis, and different demographic and public health related variables on the y axis.** Counties are sized by the total number of votes made in the 2016 election, and they are colored by the 2012 and 2016 outcomes. The top row includes variables frequently discussed in the narrative around the electoral shift in these states, including the percentage of Non-Hispanic Whites in the county. There is a clear relationship between the obesity rate, physical inactivity rate, smoking rate, and life expectancy and the Republican margin shift in these states.

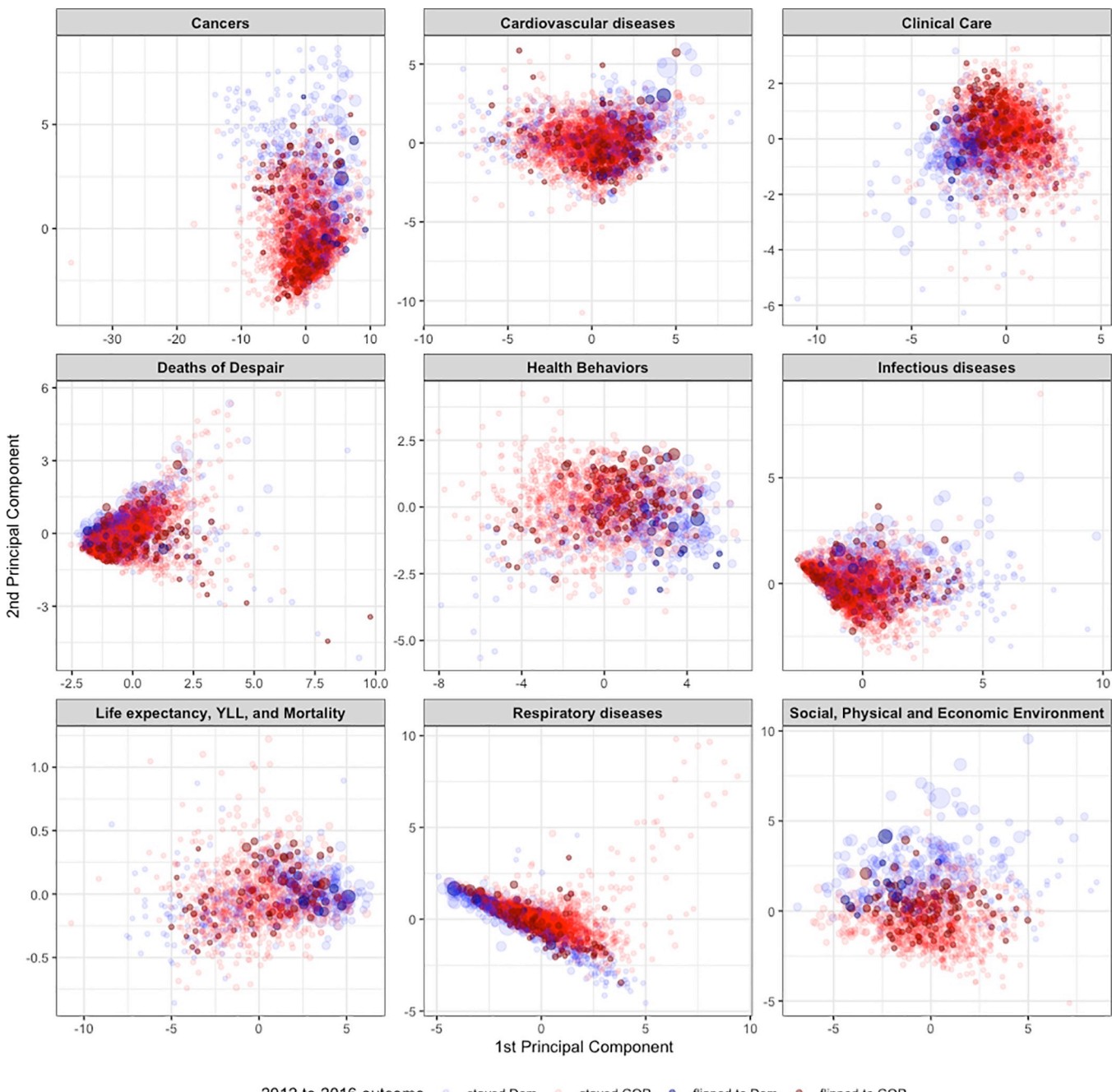

**Fig 4. Plot shows the first 2 principal components for 9 different categories of public health related variables.** Counties are sized by the total number of votes made in the 2016 election, and they are colored by the 2012 and 2016 outcomes.

with the Republican margin shift in flipped states include mortality risk across all age groups, the percentage of people in the counties on Medicaid across all ages/sex groups, and the overall uninsured rate.

The median Republican county had 11% fewer residents who completed some college and the bottom and top quartiles were 6% and 11% less respectively. Table 2 shows percent differences between the 1st quartile, median, 3rd quartile and means for different health-related

**Table 3. For every category, we applied lasso regression to predict the percentage of voters in the county that voted for Donald Trump or Hillary Clinton, and the Republican margin shift.**

| Prediction | Variable | Category | Rank in category | Overall coefficient |
|---|---|---|---|---|
| Rep. margin change | 'Malignant skin melanoma' | Cancers | 1 | 0.02653 |
| Rep. margin change | 'Colon & rectum cancer' | Cancers | 2 | 0.02544 |
| Rep. margin change | 'Lip & oral cavity cancer' | Cancers | 3 | 0.01959 |
| Rep. margin change | 'Tracheal, bronchus, & lung ' | Cancers | 4 | 0.01902 |
| Rep. margin change | 'Multiple myeloma' | Cancers | 5 | 0.01751 |
| Rep. margin change | 'Rheumatic heart disease' | Cardiovascular diseases | 1 | 0.02951 |
| Rep. margin change | 'Aortic aneurysm' | Cardiovascular diseases | 2 | 0.02736 |
| Rep. margin change | 'Ischemic heart disease' | Cardiovascular diseases | 3 | 0.01447 |
| Rep. margin change | 'Ischemic stroke' | Cardiovascular diseases | 4 | 0.01022 |
| Rep. margin change | Endocarditis | Cardiovascular diseases | 5 | 0.00712 |
| Rep. margin change | '% Screened' | Clinical Care | 1 | 0.01589 |
| Rep. margin change | 'PCP Rate' | Clinical Care | 2 | 0.01524 |
| Rep. margin change | 'Preventable Hosp. Rate' | Clinical Care | 3 | 0.00636 |
| Rep. margin change | '% With Access' | Clinical Care | 4 | 0.00426 |
| Rep. margin change | '% Vaccinated' | Clinical Care | 5 | 0.00341 |
| Rep. margin change | 'Self-harm' | Deaths of Despair | 1 | 0.02214 |
| Rep. margin change | 'Alcohol use disorders' | Deaths of Despair | 2 | 0.01750 |
| Rep. margin change | 'Drug use disorders' | Deaths of Despair | 3 | 0.00257 |
| Rep. margin change | 'Interpersonal violence' | Deaths of Despair | 4 | 0.00142 |
| Rep. margin change | '% Food Insecure' | Health Behaviors | 1 | 0.03853 |
| Rep. margin change | '% Smokers' | Health Behaviors | 2 | 0.03674 |
| Rep. margin change | '% Excessive Drinking' | Health Behaviors | 3 | 0.03035 |
| Rep. margin change | 'Food Environment Index' | Health Behaviors | 4 | 0.01850 |
| Rep. margin change | opioid_prescribing_rate | Health Behaviors | 5 | 0.01282 |
| Rep. margin change | Meningitis | Infectious diseases | 1 | 0.03231 |
| Rep. margin change | Hepatitis | Infectious diseases | 2 | 0.02039 |
| Rep. margin change | 'Diarrheal diseases' | Infectious diseases | 3 | 0.01853 |
| Rep. margin change | 'HIV AIDS' | Infectious diseases | 4 | 0.01534 |
| Rep. margin change | 'Lower respiratory infections' | Infectious diseases | 5 | 0.00370 |
| Rep. margin change | 'Mortality risk, age 45–65' | Life expectancy, YLL, and Mortality | 1 | 0.03939 |
| Rep. margin change | 'Mortality risk, age 25–45' | Life expectancy, YLL, and Mortality | 2 | 0.03855 |
| Rep. margin change | 'Years of Potential Life Lost Rate' | Life expectancy, YLL, and Mortality | 3 | 0.01646 |
| Rep. margin change | 'Mortality risk, age 0–5' | Life expectancy, YLL, and Mortality | 4 | 0.01298 |
| Rep. margin change | 'Age-Adjusted Mortality' | Life expectancy, YLL, and Mortality | 5 | 0.01292 |
| Rep. margin change | 'Other pneumoconiosis' | Respiratory diseases | 1 | 0.02393 |
| Rep. margin change | Asthma | Respiratory diseases | 2 | 0.01047 |
| Rep. margin change | 'Other chronic respiratory ' | Respiratory diseases | 3 | 0.00917 |
| Rep. margin change | Asbestosis | Respiratory diseases | 4 | 0.00529 |
| Rep. margin change | 'Interstitial lung disease' | Respiratory diseases | 5 | 0.00407 |
| Rep. margin change | '% Single-Parent Households' | Social, Physical and Economic Environment | 1 | 0.05328 |
| Rep. margin change | 'Firearm Fatalities Rate' | Social, Physical and Economic Environment | 2 | 0.03071 |
| Rep. margin change | '% Children in Poverty' | Social, Physical and Economic Environment | 3 | 0.01253 |
| Rep. margin change | 'Injury Death Rate' | Social, Physical and Economic Environment | 4 | 0.00925 |
| Rep. margin change | '% Homeowners' | Social, Physical and Economic Environment | 5 | 0.00901 |
| % Trump 2016 | 'Acute lymphoid leukemia' | Cancers | 1 | 0.03711 |
| % Trump 2016 | 'Liver cancer' | Cancers | 2 | 0.02191 |

*(Continued)*

**Table 3.** (Continued)

| Prediction | Variable | Category | Rank in category | Overall coefficient |
|---|---|---|---|---|
| % Trump 2016 | 'Other pharynx cancer' | Cancers | 3 | 0.01915 |
| % Trump 2016 | 'Nasopharynx cancer' | Cancers | 4 | 0.01814 |
| % Trump 2016 | 'Kidney cancer' | Cancers | 5 | 0.01802 |
| % Trump 2016 | 'Other cardiovascular' | Cardiovascular diseases | 1 | 0.01340 |
| % Trump 2016 | 'Ischemic heart disease' | Cardiovascular diseases | 2 | 0.01295 |
| % Trump 2016 | 'Rheumatic heart disease' | Cardiovascular diseases | 3 | 0.01012 |
| % Trump 2016 | 'Aortic aneurysm' | Cardiovascular diseases | 4 | 0.00909 |
| % Trump 2016 | 'Ischemic stroke' | Cardiovascular diseases | 5 | 0.00889 |
| % Trump 2016 | '% Screened' | Clinical Care | 1 | 0.02656 |
| % Trump 2016 | 'MHP Rate' | Clinical Care | 2 | 0.02017 |
| % Trump 2016 | '% Vaccinated' | Clinical Care | 3 | 0.01250 |
| % Trump 2016 | '% With Access' | Clinical Care | 4 | 0.00803 |
| % Trump 2016 | 'PCP Rate' | Clinical Care | 5 | 0.00722 |
| % Trump 2016 | 'Self-harm' | Deaths of Despair | 1 | 0.03750 |
| % Trump 2016 | 'Alcohol use disorders' | Deaths of Despair | 2 | 0.03498 |
| % Trump 2016 | 'Interpersonal violence' | Deaths of Despair | 3 | 0.02354 |
| % Trump 2016 | 'Drug use disorders' | Deaths of Despair | 4 | 0.00690 |
| % Trump 2016 | physical_inactivity_crude | Health Behaviors | 1 | 0.03454 |
| % Trump 2016 | '% Food Insecure' | Health Behaviors | 2 | 0.02989 |
| % Trump 2016 | 'MV Mortality Rate' | Health Behaviors | 3 | 0.02536 |
| % Trump 2016 | 'Food Environment Index' | Health Behaviors | 4 | 0.02194 |
| % Trump 2016 | 'Teen Birth Rate' | Health Behaviors | 5 | 0.02165 |
| % Trump 2016 | Hepatitis | Infectious diseases | 1 | 0.03141 |
| % Trump 2016 | 'Diarrheal diseases' | Infectious diseases | 2 | 0.02719 |
| % Trump 2016 | 'HIV AIDS' | Infectious diseases | 3 | 0.02079 |
| % Trump 2016 | 'Lower respiratory infections' | Infectious diseases | 4 | 0.01920 |
| % Trump 2016 | Meningitis | Infectious diseases | 5 | 0.00957 |
| % Trump 2016 | 'Years of Potential Life Lost Rate' | Life expectancy, YLL, and Mortality | 1 | 0.09069 |
| % Trump 2016 | 'Mortality risk, age 5–25' | Life expectancy, YLL, and Mortality | 2 | 0.07097 |
| % Trump 2016 | 'YPLL Rate (White)' | Life expectancy, YLL, and Mortality | 3 | 0.04372 |
| % Trump 2016 | 'Mortality risk, age 0–5' | Life expectancy, YLL, and Mortality | 4 | 0.02479 |
| % Trump 2016 | 'Mortality risk, age 65–85' | Life expectancy, YLL, and Mortality | 5 | 0.02457 |
| % Trump 2016 | 'Chronic obstructive pulmonary ' | Respiratory diseases | 1 | 0.04661 |
| % Trump 2016 | 'Interstitial lung disease' | Respiratory diseases | 2 | 0.01623 |
| % Trump 2016 | Asthma | Respiratory diseases | 3 | 0.01243 |
| % Trump 2016 | 'Other pneumoconiosis' | Respiratory diseases | 4 | 0.00720 |
| % Trump 2016 | 'Other chronic respiratory ' | Respiratory diseases | 5 | 0.00712 |
| % Trump 2016 | '% Single-Parent Households' | Social, Physical and Economic Environment | 1 | 0.06653 |
| % Trump 2016 | '% Severe Housing Problems' | Social, Physical and Economic Environment | 2 | 0.03953 |
| % Trump 2016 | 'Firearm Fatalities Rate' | Social, Physical and Economic Environment | 3 | 0.03877 |
| % Trump 2016 | '% Homeowners' | Social, Physical and Economic Environment | 4 | 0.01254 |
| % Trump 2016 | '% Children in Poverty' | Social, Physical and Economic Environment | 5 | 0.01000 |

This table reports the variable importance for the top variables in each category.

variables as well as t-statistics and p-values. The median Republican county had a 17% higher "injury death rate" and a 26% higher "% Disconnected Youth" rate. The median Republican county had a 50% higher rate of coal workers' pneumoconiosis, 25% higher rate of chronic

respiratory diseases, and a 32% higher rate of chronic obstructive pulmonary disease. The mortality risk was higher in every age group for the median and bottom quartile counties, and lower for the age groups less than 65. The healthcare costs per capita were higher in Democratic counties (including Imaging Costs per Capita, Procedures Per Capita Actual Costs, Tests Per Capita Actual Costs, and Actual Costs per capita). There were consistently higher Medicaid participation rates in Democratic counties. The median uninsured % and uninsured % among individuals with less than 138% of the poverty line were higher in Republican counties by 2% and 11%, respectively. The insurance rates were much higher in Democratic counties for those with and without a high school education. The median Republican county had 2% lower life expectancy overall and 3% lower life expectancy among whites. Cancer rates were higher in most Republican counties than Democratic counties, with the exceptions of prostate, liver, and stomach cancers.

The median Republican county had a 13% higher obesity rate, a 21% higher diabetes rate, a 19% higher physical inactivity rate, a 24% higher opioid prescribing rate, and a 6% higher smoking rate. Republican counties are older, with the median Republican county having 21% more individuals in the % 65 and over demographic. They are also whiter, with the median Republican county having a 69% greater rate in the % Non-Hispanic White demographic. Republican counties are more rural (median % rural rate is 234% higher for Republican counties), and access to care decreases in these counties accordingly: the primary care physician rate (ratio of population to primary care physicians) was 37% lower in the median Republican counties. Some of these health behavior, life expectancy, and health insurance rate differences presented in Table 2 are visualized in Fig 1.

Fig 2 shows the dynamics of healthcare and mortality in Democratic and Republican counties over time, visualizing the rates of different diseases and mortality over time. Over the past 10+ years, life expectancy has changed at different rates, and has improved faster in Democratic counties. Since 2008, health behavior measures and chronic diseases such as physical inactivity, diabetes, and obesity have become notably worse in Republican counties. While the mortality risk across all age groups has decreased overall since 1980, the mortality risk is now higher in the median Republican county compared to the median Democratic county for all age groups. S1 Fig clearly shows the growing differences between several health and life expectancy measures in the median Republican and Democratic counties over time.

Much of the 2016 election media narrative was focused on the rural, white, over 65 voters who supported Trump. Fig 3 shows health and demographic variables that are strongly correlated with the Republican margin shift in counties in the states that flipped from Democratic to Republican in 2016. We can also see that obesity, diabetes, physical inactivity and smoking are all highly correlated with the Republican margin shift. There is a strong negative correlation between the life expectancy of whites and the Republican margin shift in these flipped states.

S4 Table, reports coefficients for each variable from a multivariate linear model that includes each variable as well as education, socio-economic and demographic control variables (there are strong correlations between health, education, socioeconomic status and county demographics, and it is important that we include these and other demographic variables in our model). Among public health variables with the biggest positive coefficient for predicting the Republican Margin change are Medicaid variables across different age groups and sexes, the percentage of smokers, the percentage of excessive drinkers, and the obesity rate; while some of the largest negative coefficients were the uninsured rates, malignant skin melanoma, and Part B Drugs Actual Costs. Several of the health behavior variables that were highly correlated with the voting outcomes in univariate models had directional changes when we added control variables.

S2 Fig shows a plot of the correlation matrix clustered for a selection of public health and control variables. Fig 4 shows the first 2 principal components of different public health categories plotted for counties; there is a clear pattern of counties clustering based on their Republican margin shift and the percentage of people that voted for Trump across every public health category. The clustering is striking for many of the categories, especially Health Behaviors, Clinical Care, Cancers, and Life expectancy, Years of Life Lost, and Mortality. The first 2 principal components explain 48% of the variance of the dataset. The variables most correlated with the first principal component are mortality related variables, including mortality risk, age 45–65, age-adjusted mortality, years of potential life lost rate, mortality risk, age 25–45, mortality risk, age 0–5, and age-adjusted mortality (white). The variables that are most correlated with the second PC include cancer-related variables, such as chronic lymphoid leukemia, malignant skin melanoma, leukemia, non-Hodgkin lymphoma. Table 3 shows the importance of the different variables in each category when they are all included in a lasso regression model to predict voting in the county, along with control variables. The most important variables when predicting the percentage that voted for Trump among clinical care variables was the % Screened, and the MHP Rate; among Deaths of Despair variables was the self-harm rate, and the alcohol use disorder rate; among Health Behaviors was the physical inactivity rate, among the life expectancy, YLL, and Mortality category, was 'Years of Potential Life Lost Rate'-and 'Mortality risk, age 5–25', and among Respiratory diseases was 'Chronic obstructive pulmonary'. Noticeable differences in the important variables when predicting the Republican margin change were the 'Malignant skin melanoma'among Cancers, the PCP Rate among Clinical Care, the % Food Insecure and '% Smokers'among Health Behaviors, and the 'Mortality risk, age 45–65', and 'Mortality risk, age 25–45'among Life expectancy, YLL, and Mortality variables.

Most of the states that participated in Medicaid expansion (though not all) voted Democratic in 2016. States that expanded Medicaid improved the insurance rates of their states and tended to have higher insured rate changes than states that did not, although there are a few exceptions (such as Florida and Idaho) that experienced large changes over this period without expansion. Medicaid expansion was particularly impactful on the insurance rates of individuals making less than 138% of the poverty line. S3 Fig shows insurance rate changes from 2008 to 2017 (capturing the impact of the ACA) for counties in States that did and did not implement Medicaid expansion. Each point represents a county. For this group, Medicaid expansion directly improved the insurance rates of states.

## Discussion

In this retrospective cohort study, we found statistically significant relationships between a number of health measures and the political voting patterns of counties in 2016 and over the last three decades. By calculating the median difference between counties that voted Democratic or Republican in the 2016 election, we found that residents of counties that voted Republican in the last presidential election had increased median incidence cardiovascular disease (11% median difference), diabetes (21%), obesity (13%), self-harm (22%), decreased median life expectancy (2%), and physical activity (19%) compared to residents of counties that voted Democrat. Collectively, these data indicates that counties that voted Republican in the 2016 election had very different health outcomes than those that voted Democratic, and generally had a greater proportion of their residents in poor health.

Fig 2 shows that counties that voted Republican in 2016 had increases in negative health outcomes such as diabetes and obesity concomitantly with decreases in life expectancy compared to counties that voted Democratic in 2016. This indicates that these counties have

experienced an overall worsening in quality of health over time. It is important to note that these are not necessarily counties that have voted Republican in previous elections.

We have examined the relationship between state voting outcome and Medicaid expansion. We showed that states that expanded Medicaid (at the time of this writing) were more likely to be Democrat and had improved health measures, including improved access to care, better glucose monitoring in diabetes, better hypertension control, reductions in rates of major post-operative morbidity, and reductions in preventable hospitalizations, compared to those that did not [38–40]. The 14 states that have not expanded Medicaid (Alabama, Florida, Georgia, Kansas, Mississippi, Missouri, North Carolina, Oklahoma, South Carolina, South Dakota, Tennessee, Texas, Wisconsin, Wyoming) have overall lower median insured rates among those making 138% below the federal poverty level compared to states that expanded Medicaid. In future research, it will be important to study patterns between health policy and political party affiliation more comprehensively, as health policies have a direct impact on public health.

Appropriate resource allocation is a crucial driver of healthcare outcomes and we hope that the data presented can be used to guide policy decisions at a tractable level. 52% of Medicare funds are allocated to counties that voted Republican in 2016 and 55.5% of individuals with preexisting conditions live in states that voted Republican that same year. These data indicate that, contrary to the policies of the Trump administration and attempts by Republican legislators to cut Medicare and other entitlement spending [41], these programs should actually be expanded and better targeted to counties that require them. For example, programs aimed at improving access to care and adherence to treatment in older patients with chronic illnesses (e.g. diabetes or chronic obstructive pulmonary disease) who live in specific rural communities would require a different allocation of resources, but has the potential to decrease morbidity and mortality. If the $237 billion annual healthcare expenditure for diabetes were directed more pointedly towards prevention, screening, and optimizing treatment, it is likely outcomes would improve and spending will decrease over time as the disease is caught earlier and treated more effectively before it can cause major morbidity.

The limitations of this study are as follows. First, we cannot attribute voting behavior to individual Democratic and Republican voters. Voter turnout in the United States is low (55.7% in the 2016 election) compared to other developed countries and varies between demographics. Therefore, the available data do not allow us the infer whether the results in a given county reflect the true preferences of its residents. Similarly, the majority of lower socioeconomic individuals do not vote, which skews the data towards those who can, who may also be healthier on average. As such, these data do not indicate the healthcare differences between individual voters and should not be understood as a reflection of individual party member preferences. Similarly, these data does not adequately account for independent and third-party voters.

There are additional healthcare access variables besides insurance rates that we do not include in this study. While we include access variables like the PCP rate, we cannot completely capture the quality and breadth of healthcare available in individual counties. For example, the number of healthcare facilities, the training of healthcare personnel, and the quality of both will vary widely. This distinction is especially relevant between rural and urban healthcare settings. Because we cannot adequately control for the quantity and quality of healthcare and access, we cannot conclude that Republican counties would have worse outcomes if they had the same resources. Republican counties tend to be much more rural, leading to demographic differences. As younger people tend to move from rural to densely populated areas, the age makeup of Republican and Democratic counties will also vary, affecting the health rates of those counties.

As our multivariate analysis found, the public health of counties is strongly correlated with the education level, socio-economic status and demographics of counties, making it hard to quantify the independent relationship between each public health variable and voting. Any multivariate analysis with highly correlated variables can be very fragile, and this is certainly true in our case. As previously mentioned, counties that voted Democratic in the 2020 election accounted for 70% of the US GDP. The socioeconomic status of counties will have a major impact on both healthcare access and the health of those living in those counties.

We found strong relationships between recent county voting patterns and health outcomes. These outcomes stem from both individual mechanisms (like the aforementioned lower priority of health issues of Republican voters) as well as institutional aggregate measures (e.g. ACA and Medicare expansion choices falling along party lines). Polarization and partisanship are increasing in the US, and our work suggests that it is in the public interest to further study the mechanisms that link partisanship to health outcomes in an attempt to decouple political affiliation and health in the future.

## Supporting information

**S1 Table. Pearson correlations between all of the public health-related variables we collected with the percentage of voters in the county that voted for Donald Trump or Hillary Clinton, and the Republican margin shift (from 2012 to 2016).** Correlations for counties from all states, counties from battleground states, and counties from states that flipped from Democratic in 2012 to Republican in 2016 are presented.
(DOCX)

**S2 Table. Weighted Pearson correlations (weighted by the log 10 of the county population) between all of the public health-related variables we collected with the percentage of voters in the county that voted for Donald Trump or Hillary Clinton, and the Republican margin shift (from 2012 to 2016).**
(DOCX)

**S3 Table. Quantile and mean comparisons of Republican and Democratic counties across all of the public-health measures we collected.** Every county was assigned as either Republican or Democratic depending on the majority vote in 2016, and the mean, median, 1st quartile, and 3rd quartile values for different public health-related variables were calculated. The differences in these values for Republican and Democratic counties are presented in S3 Table, along with the Student t-test statistics and p values for the mean comparisons.
(DOCX)

**S4 Table. This table reports the coefficients for each public health variable under consideration, as well as the standard error for those coefficients, when predicting the percentage of voters in the county that voted for Donald Trump or Hillary Clinton, and the Republican margin shift.** Each linear model included education, socio-economic status, and demographic control variables for the county.
(DOCX)

**S1 Fig. Percent differences in the median value of Republican and Democratic counties for select life expectancy, mortality, and health behavior measures.** The median Republican county has experienced sustained increases in mortality risk across every age group compared to the median Democratic county between 1980 and 2014; this manifests itself in worse life expectancy for the median Republican counties over time. Diabetes, obesity, physical inactivity, and uninsurance rates in the median Republican counties are higher than in the median

Democratic counties between 2006 and 2017, and this difference is growing.
(EPS)

**S2 Fig. This is a clustered plot of the correlation matrix for a select group of public health, education, socio-economic and demographic variables.**
(TIFF)

**S3 Fig. Boxplots of insurance rate changes between 2008 and 2017 for counties in states.**
Boxplots are filled by whether the state expanded Medicaid, and state names are colored by the 2016 political party. States that expanded Medicaid experienced higher insurance rate changes during this time period, indicating the positive impact of the policy.
(EPS)

## Acknowledgments

We thank Kevin Aslett for discussions and a careful reading of the manuscript.

## Author Contributions

**Conceptualization:** Tymor Hamamsy, Michael Danziger, Jonathan Nagler, Richard Bonneau.

**Data curation:** Tymor Hamamsy.

**Formal analysis:** Tymor Hamamsy.

**Funding acquisition:** Richard Bonneau.

**Investigation:** Tymor Hamamsy, Michael Danziger, Jonathan Nagler, Richard Bonneau.

**Methodology:** Tymor Hamamsy, Michael Danziger, Jonathan Nagler, Richard Bonneau.

**Project administration:** Richard Bonneau.

**Resources:** Richard Bonneau.

**Software:** Tymor Hamamsy.

**Supervision:** Jonathan Nagler, Richard Bonneau.

**Validation:** Tymor Hamamsy, Jonathan Nagler.

**Visualization:** Tymor Hamamsy, Michael Danziger, Jonathan Nagler.

**Writing – original draft:** Tymor Hamamsy, Michael Danziger, Richard Bonneau.

**Writing – review & editing:** Tymor Hamamsy, Michael Danziger, Jonathan Nagler, Richard Bonneau.

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
