## [Decision Letter · Decision Letter 0]

3 Dec 2020

PONE-D-20-17485

Viewing the US presidential electoral map through the lens of public health.

PLOS ONE

Dear Dr. Hamamsy,

Thank you for submitting your manuscript to PLOS ONE. After careful consideration, we feel that it has merit but does not fully meet PLOS ONE’s publication criteria as it currently stands. Therefore, we invite you to submit a revised version of the manuscript that addresses the points raised during the review process.

Please revise the manuscript incorporating recommendations by reviewers 1 and 2. Reviewer 1 states that there is no literature review and that a clear hypothesis or purpose of the paper appears to be lacking. Please conduct a literature review and state your hypothesis clearly. Also please address the issues of statistical concern to reviewer 1. Please also explain why socioeconomic status is often more important than higher insurance rates.

Please follow closely recommendations by reviewer 2, especially when terms of definition are lacking or unclear. Also please define your political affiliation variable considerably earlier than page 5. Please revise in particular the summary statement at the end of the abstract which reviewer 2 finds highly problematic. In addition, please add a brief discussion of additional limitations to the study, as suggested by reviewer 2. As reviewer 2 suggests, please remove the material in pages 45-48, as this material starts a second a research article. You can make reference to this material as issues to consider in a follow-up study.

We look forward to receiving your revised manuscript.

Kind regards,

M. Harvey Brenner, PhD

Academic Editor

PLOS ONE

Journal Requirements:

"I have read the journal's policy and the authors of this manuscript have the following competing interests: R.B. has ongoing or recent consulting or advisory relationships with Eli Lily, Merus, Merck and Epistemic AI. R.B. has an active research collaboration with Facebook. TH cofounded Fermat’s Library. "

Reviewers' comments:

Reviewer's Responses to Questions

**Comments to the Author**

1. Is the manuscript technically sound, and do the data support the conclusions?

Reviewer #1: Partly

Reviewer #2: Yes

2. Has the statistical analysis been performed appropriately and rigorously? 

Reviewer #1: Yes

Reviewer #2: Yes

3. Have the authors made all data underlying the findings in their manuscript fully available?

Reviewer #1: Yes

Reviewer #2: Yes

4. Is the manuscript presented in an intelligible fashion and written in standard English?

Reviewer #1: No

Reviewer #2: No

5. Review Comments to the Author

Reviewer #1: This is an interesting and important study. But it needs much more work. Here are my concerns:

There is no literature review. Many of the references are newspaper or other non-scholarly sources. A literature review of peer-reviewed sources is necessary to establish the contribution of this paper. My guess is that many observers are aware that Trump's support is concentrated in rural areas and among white men of lower socio-economic status. These are some of you most robust findings. But what may not be understood is the distribution and prevalence of specific health conditions among Trump supporters. You find a high prevalence of melanoma and COPD for example. I did not know this. Such a focus would, I suspect, be a contribution to the literature but this should be established.

There is no clear hypothesis or purpose of the paper. This could be improved with a better sense of how this work contributes to the literature.

In its present form, the paper seems a bit of a data dump. Much of the data shown in the paper must be pared back. I suggest limiting Table 1 to the most important findings. The remainder can go to an appendix. The same for Table 2. In fact, perhaps Table 2 could be eliminated and discussed in several paragraphs.

The statistical analysis concerns me. A more rigorous approach would employ use of factor analysis or principle component analysis to determine which groups of variables are most important. Use of correlations may be adequate but a statistical analysis of those correlations would be better. I strongly suggest considering further multivariate work.

It does seem a bit odd that so many Trump supporters are in relatively ill health and dependent on public finance of health services yet oppose expansion and other aspects of Medicare, Medicaid and the like. You offer no explanation of this. I think some speculation is called for. Why do they oppose policies that are seemingly in their interest? Perhaps rejection of the "elites" is more important or maybe a sense of independence and self dignity, in spite of material circumstances explains the findings. I certainly don't know but some further exploration should be strongly considered. This might help establish a better understanding of the relationship between health and voting patterns.

You also show that higher insurance rates do not necessarily improve health. This is important and should be underscored. Socio-economic status is often more important. Why? Explain.

Reviewer #2: The authors are examining the relationship of county-level data on morbidity, disability, mortality, life expectancy, health care insurance, and other community social and well-being indicators to data on the county-level political voting outcomes in the presidential elections 2012 and 2016. Unfortunately, the authors do not explain why they chose to work with county-level data, other than the reference to geographical trends in voting patterns. Yet, many readers located without and within the USA may lack understanding of a county's role in government, election boards , registration of political affiliation, certification of election results, -- or even the distinction between the popular and the electoral college vote. Readers will better appreciate this research if the authors provide more detail about their choice to focus on county level data from a public health perspective.

The authors create a problem by waiting until the Methods and Materials section, page 5 to define their county political affiliation variable. There we learn that a county is classified as Democrat or Republican depending upon which political party garnered the greater number of popular votes in favor of its presidential candidate in that election year, But we do not learn this until page 5, so many passages in the abstract and introductory text, referring to Republican or Democrat states/districts/counties/voters , can be confusing before getting to the Methods section. Examples of ambiguous or confusing passages follow:

Paragraph 1 of the Introduction cites articles on the priorities of voters registered as Republican or Democrat -- omits registration as Independent -- and refers to them as 'Republican and Democratic voters'. Do the authors consider the words 'Democrat' and 'Democratic' interchangeable?

Paragraph 2 goes on to cite published data on median household income differences between Republican and Democratic districts. Can we assume the authors mean the median income in districts where in some unspecified election year the Republican (or Democrat) candidate won? Or. do the authors mean that these are districts where the election boards have a greater number of citizens registering as Republican (or Democrat) than those registering with another political affiliation? Or, do they mean the unlikely possibility that the income level and political preference is known for each person who votes in an election?

The paragraph next states 'Republican states' have experienced relative wage stagnation. Are these 'Republican states' those with Republican governors? Or, are they ones with a Republican majority in the state Senate? Or, do they have more residents registered as Republicans than those registered with a different party affiliation.? Or, did that state's delegate to the electoral college vote for the Republican presidential candidate in the most recent election year? Perhaps a Republican candidate for president won the majority of a state's popular votes in a recent election? It just is not clear.

The third paragraph of the Introduction states the major causes of death show 'geographical trends'. The word 'geographical has a wide range of meaning, What is meant by this study? Is a state categorized by the extent to which its territory is composed of mountains, plains, dry deserts, forests, lakes, or with coastline? Does 'geographical' refer to how the state is populated, e.g. urban, peri-urban, small town, rural? Or, are states categorized by location and direction, e.g., Border states. East Coast, West Coast, Middle, Upper Middle, South, North, or by some other 'geographical' distinction?

The authors sometimes use terms that might need more explanation. For example, page 3, the authors say "It has been shown that Trump over-performed in counties with high drug. alcohol. and suicide rates." Is there wide-spread understanding of the phrase 'over-performed'? A similar question could be raised about the term 'battleground states' (see Introduction paragraph 4), though the authors do define this term further on in the Methods section.

There is an egregious summary statement at the end of the abstract, and repeated twice near the end of the article, but by that time there has been sufficient explanation that there is less chance for the reader to misinterpret. The offending but cogent statement is "Collectively, this data exhibits a strong pattern: counties that voted Republican in the 2016 election are 'sicker' than those that voted Democrat." The authors are referring to the finding that counties where the majority of votes were for the Republican presidential candidate, were also the counties where a higher percentage of residents were in poor physical health, by a number of public health measures, compared to counties where the candidate of the Democrats won the majority vote. (Whether the ailing residents actually voted and who they voted for, we do not know.)

The authors must be aware that another, more pejorative meaning of 'sicker' is "morally unsound or corrupt" (Webster). If this article is accepted, it will be published after a hotly contested national presidential election where weeks later emotions continue to run high and misinterpretations can feed the existing political' and cultural fracture. The summary statement of the authors is a headline=grabbing description of this study's collective findings, but it does not belong in a scientific article because of the possible derogatory interpretation of a cultural and political group. when more accurate and precise wording is possible.

Indeed, the authors do clearly define their variables in the Methods section and appropriately label scatter plots and the supplemental box plot figures in the Analysis section of the article. (Note also that the authors provide a comprehensive list of relevant references and available sources of data used in this research,.) But, by waiting until page 5 in the manuscript to define their terms, the authors allow misunderstanding to take root in the Abstract and introductory sections so that it is possible for someone to quote a passage out of context and undercut belief in an unbiased science. A good copy editor should be able to flag all these instances where the reader could misinterpret the terms and associated results, and improve clarity.

In general, the ample data analysis in this study supports the conclusion that those counties where the Republican presidential candidate won a majority of the popular vote (i.e. Republican counties) are counties that have a greater proportion of residents considered to be ailing or in poor health, and not doing as well economically as residents of those counties where the candidate of the Democrats garnered the majority of votes for President. This study is very complete in showing the correlations between a large number of a health and community well-being variables commonly used in the public health field and the voting patters of the two main political parties. It is the consistency of patterns in those collective findings that makes the study's finding convincing.

To their credit, the research team took two additional steps in analysis that enhance the readers' understanding of the relationship between voting patterns and health status at the county level. The first additional step was to examine the data for discontinuities as revealed by a positive or negative shift in in the majority votes for a political party's presidential candidate, 2016-2012. The second step was to examine trends in a county's well-being and health variables over time. This addresses questions of whether conditions in the county are better now or worse than before, Is there more or less of a particular dynamic now, and if so, how does that trend relate to voting patterns in the presidential election?

The authors do cite some limitations on this research study, but could further strengthen this article by adding a brief discussion about two other possible limitations on the interpretations of their results, specifically, acknowledging the possible role of additional health care access variables other than insurance, and the influence that demographic change in rural areas and small towns of the USA has on the assessment of health status. Both could influence interpretations drawn from the data analysis.

In the USA, over time, more and more people 20-39yrs have moved away from small towns and rural areas, leaving older family members, and fewer younger families and children remaining in the area. There has been little replacement, or in-migration to offset the county's loss. A population with this changed age structure has more chronic illness than a population that maintains a more normal age distribution. Such demographic change has been accompanied by decaying downtown business areas, fewer jobs, and less innovation in those same small towns and rural areas, changes which breed deaths of despair. Drug addiction, and suicide are responses to less economic hope and few support structures. Is it possible that the finding that those voting for the Republican presidential candidate are 'sicker', is just a twist on the well-known fact that Trump won the vote of older, conservative, and rural Americans in 2016 and were chiefly responsible for his success in the electoral college vote?

The last decade has produced many public health studies demonstrating the importance of health care access to the health of an individual and a community. Health care costs and insurance to offset those costs are an important part of access, The study team wisely included a health care insurance variable in their analysis and the authors detail those findings in the article . But, ability to pay is not the sole determinant of health care access. The actual presence of facilities and trained health personnel are necessary to provide not only emergency, acute and chronic medical care for physical and mental health crises, but also rehabilitation for stroke, heart attack, fracture, and trauma -- problems frequently seen in counties with an older, rural, or small own population, and decaying infrastructure. However, in recent news reports, we have learned of the closing of more hospitals that provide services to small towns and rural areas, Not only facilities are few in number, fewer trained health professionals are are located there. Nurses, health educators, primary care providers, as well as those in geriatrics, mental health, pulmonology and other medical specialties are mainly located in or near urban areas with large populations.

The authors point out that 'Republican counties' have a higher proportion of residents with behavioral health problems than in 'Democratic counties'. Behavioral health problems can by impacted by access to education. counseling, group support, mentoring practices, and program sponsored rewards and other incentives. Facilities and trained health professionals are needed to promote behavioral health through classes on stopping smoking, weight management, diabetes control and nutrition, stress management, balance and strength raining, grief support, and small group activities designed to foster interchange and reduce social isolation. Yet small towns rand rural areas have limited access to community health or social programs sponsored by health facilities, non-profits, local agencies university or government. Transport itself can be a problem depending on how far away a program is being held. No wonder the emphasis on self-reliance and do-it-yourself. Who and what is there to be of help to most small town and rural folk? The answer is kin (what there is left of them), neighbors and the church.

The question remains: Would the 'Republican counties have a 'sicker' population with more risk factors, (e.g. smoking, obesity, and lack of physical activity), if they had access to the same or similar health care resources as the counties where a Democrat presidential candidate won the the majority vote? Such additional access variables as presence and number of health care facilities, trained health personnel, and transport time to care were outside the focus of this particular research study, but are possible limitation on interpretation of study findings.

The final pages of text pages 45-48, begin to explore a fascinating topic--patterns between health policy and political party affiliation, and the current toxic influence of partisanship on health funding, practices, and outcomes, including a short discussion of issues around the 2020 coronavirus pandemic. In effect, the authors start a second, but connected research question and discussion. What the authors have to say is very worthwhile and needs to be pursued as another research article, and draft article detailing question, methods, analysis and conclusions -- or possibly adapt for an Op-Ed in a major national newspaper. I hope the authors will pursue this inquiry. They have the interest, background knowledge, and have done the groundwork. But, the topic and existing associated text should be considered a discrete inquiry and discussion, separate from this article on the relationship of a county population's health and social well-being and their voting patterns in 2012 and 2016 presidential election.

Recommendation: Accept with revisions.

6. PLOS authors have the option to publish the peer review history of their article (what does this mean?). If published, this will include your full peer review and any attached files.

Reviewer #1: **Yes: **Peter Hilsenrath

Reviewer #2: **Yes: **Dory Storms, ScD, MPH

---

## [Author Response · Author response to Decision Letter 0]

12 Jan 2021

Dear Dr. Brenner,

Thank you very much for taking the time to thoroughly review our article and for the helpful feedback. We have attempted to address all of the reviewer comments, and we very much appreciate their careful review. 

Regarding the first reviewer’s comments, we have added a more substantial literature review to the introduction, and we clearly state our hypothesis and the purpose of the paper. We have added additional methodological detail and explanations for why we take the univariate approach that we do. We have also added a multivariate analysis for investigating the relationship between public health and voting patterns, controlling for the education levels, socio-economic status, and demographics of counties. We next tried to investigate the importance of different variables and variable categories. First, we performed a principal components analysis for each category of public health variables to see if counties clustered based on voting patterns for different categories. We next applied lasso regression using all of the variables from each category, and calculated their variable importance, finally ranking variables. 

Regarding the second reviewer’s comments, we have followed them closely and addressed them all below. We have fixed the language and terms that were unclear/undefined. We now define our political affiliation variable at the beginning of the paper. We removed the problematic summary statement involving “sicker”. We substantially extended our limitation section to reflect the limitations that reviewer 2 suggested, as well as the limitations that come with our univariate approach. We have now removed the material in pages 45-48 in the discussion.

All reviewer comments are addressed in detail below.

Thanks again. 

Rich, Mike, Jonathan, and Tymor

Reviewer #1: This is an interesting and important study. But it needs much more work. Here are my concerns:

There is no literature review. Many of the references are newspaper or other non-scholarly sources. A literature review of peer-reviewed sources is necessary to establish the contribution of this paper. My guess is that many observers are aware that Trump's support is concentrated in rural areas and among white men of lower socio-economic status. These are some of you most robust findings. But what may not be understood is the distribution and prevalence of specific health conditions among Trump supporters. You find a high prevalence of melanoma and COPD for example. I did not know this. Such a focus would, I suspect, be a contribution to the literature but this should be established.

There is no clear hypothesis or purpose of the paper. This could be improved with a better sense of how this work contributes to the literature.

We introduced a more substantial literature review to the paper. There is existing literature that looks at health behavior and voting patterns, life expectancy and voting patterns, as well as deaths of despair and voting patterns. A lot of these studies are great, and our contribution, which we stress in the introduction, is a comprehensive exploratory analysis that includes a broader set of public health variables than previous studies (including COPD and other variables that have not been connected to voting patterns before), and collectively indicate worse health in Republican counties.

In its present form, the paper seems a bit of a data dump. Much of the data shown in the paper must be pared back. I suggest limiting Table 1 to the most important findings. The remainder can go to an appendix. The same for Table 2. In fact, perhaps Table 2 could be eliminated and discussed in several paragraphs.

We reduced Table 1 and Table 2 to some of the most important findings. We create 2 supplementary tables for the appendix, that are just the original Table 1 and Table 2 that include all of the 150+ variables.

Part of the purpose of the paper is to be a bit of a data dump - as we are including a very broad set of public health variables, we hope that this paper can be a resource for people, and help others generate ideas.

The statistical analysis concerns me. A more rigorous approach would employ use of factor analysis or principle component analysis to determine which groups of variables are most important. Use of correlations may be adequate but a statistical analysis of those correlations would be better. I strongly suggest considering further multivariate work.

In table 2, we do include p values and t statistics, and in table 1, all of the correlations are statistically significant- we include 95% confidence intervals.

We agree entirely about the importance of multivariate work, especially as so many of the variables in our study are correlated with one another. We introduce a visualization of the correlation table for all of the variables from table 1 and table 2, as a supplementary figure. Indeed, public health is extremely correlated with the education levels, socioeconomic status, and demographics of counties. We have added a multivariate analysis for investigating the relationship between public health and voting patterns, controlling for the education levels, socio-economic status, and demographics of counties. In order to investigate the importance of different variables and variable categories, we performed a principal components analysis for each category of public health variables, visualizing counties using the first two principal components, and showing clustering of counties based on both the Republican margin shift and the percentage of voters for Trump in the county. We next applied lasso regression using all of the variables in each category as well as the control county variables, and measured the variable importance of variables from each category, ranking the most important variables.

The main intention with this study was to comprehensively explore and describe the public health of counties through the lens of voting patterns, and we therefore do not make any strong claims/inferences from our multivariate analysis. Multivariate analysis with highly correlated variables can be very fragile, and we explain this in our limitations section. 

It does seem a bit odd that so many Trump supporters are in relatively ill health and dependent on public finance of health services yet oppose expansion and other aspects of Medicare, Medicaid and the like. You offer no explanation of this. I think some speculation is called for. Why do they oppose policies that are seemingly in their interest? Perhaps rejection of the "elites" is more important or maybe a sense of independence and self dignity, in spite of material circumstances explains the findings. I certainly don't know but some further exploration should be strongly considered. This might help establish a better understanding of the relationship between health and voting patterns.

This is a very interesting point, and there is certainly a long history of people voting against their interests in american politics, especially given the polarized media environment and misinformation on social media. As we removed pages 45-48 to address reviewer 2’s concerns, this section was also removed.

You also show that higher insurance rates do not necessarily improve health. This is important and should be underscored. Socio-economic status is often more important. Why? Explain.

Socioeconomic status is extremely important, and we add a discussion about it to our limitations section, as it is certainly a driver behind healthcare access, quality, and health behavior. 

Insurance rates do not necessarily improve health, and we discuss a host of other health access/health quality related variables in the limitation section that address this point.

Reviewer #2: The authors are examining the relationship of county-level data on morbidity, disability, mortality, life expectancy, health care insurance, and other community social and well-being indicators to data on the county-level political voting outcomes in the presidential elections 2012 and 2016. Unfortunately, the authors do not explain why they chose to work with county-level data, other than the reference to geographical trends in voting patterns. Yet, many readers located without and within the USA may lack understanding of a county's role in government, election boards , registration of political affiliation, certification of election results, -- or even the distinction between the popular and the electoral college vote. Readers will better appreciate this research if the authors provide more detail about their choice to focus on county level data from a public health perspective.

We added an explanation for what counties are and why counties are the best geographical unit for this study.

The authors create a problem by waiting until the Methods and Materials section, page 5 to define their county political affiliation variable. There we learn that a county is classified as Democrat or Republican depending upon which political party garnered the greater number of popular votes in favor of its presidential candidate in that election year, But we do not learn this until page 5, so many passages in the abstract and introductory text, referring to Republican or Democrat states/districts/counties/voters , can be confusing before getting to the Methods section. Examples of ambiguous or confusing passages follow:

Paragraph 1 of the Introduction cites articles on the priorities of voters registered as Republican or Democrat -- omits registration as Independent -- and refers to them as 'Republican and Democratic voters'. Do the authors consider the words 'Democrat' and 'Democratic' interchangeable?

Paragraph 2 goes on to cite published data on median household income differences between Republican and Democratic districts. Can we assume the authors mean the median income in districts where in some unspecified election year the Republican (or Democrat) candidate won? Or. do the authors mean that these are districts where the election boards have a greater number of citizens registering as Republican (or Democrat) than those registering with another political affiliation? Or, do they mean the unlikely possibility that the income level and political preference is known for each person who votes in an election?

The paragraph next states 'Republican states' have experienced relative wage stagnation. Are these 'Republican states' those with Republican governors? Or, are they ones with a Republican majority in the state Senate? Or, do they have more residents registered as Republicans than those registered with a different party affiliation.? Or, did that state's delegate to the electoral college vote for the Republican presidential candidate in the most recent election year? Perhaps a Republican candidate for president won the majority of a state's popular votes in a recent election? It just is not clear.

We added a definition early in the introduction for Democrat and Republilcan counties. We added clarifications for Democrat/Republican districts when they are mentioned, and we added clarifications for what is meant by Democrat and Republican states.

We consistently applied Democratic to avoid confusion, however they are interchangeable.

The third paragraph of the Introduction states the major causes of death show 'geographical trends'. The word 'geographical has a wide range of meaning, What is meant by this study? Is a state categorized by the extent to which its territory is composed of mountains, plains, dry deserts, forests, lakes, or with coastline? Does 'geographical' refer to how the state is populated, e.g. urban, peri-urban, small town, rural? Or, are states categorized by location and direction, e.g., Border states. East Coast, West Coast, Middle, Upper Middle, South, North, or by some other 'geographical' distinction?

We added an explanation for what we mean by geographical trends (regional, spatial patterns).

The authors sometimes use terms that might need more explanation. For example, page 3, the authors say "It has been shown that Trump over-performed in counties with high drug. alcohol. and suicide rates." Is there wide-spread understanding of the phrase 'over-performed'? A similar question could be raised about the term 'battleground states' (see Introduction paragraph 4), though the authors do define this term further on in the Methods section.

We now define battleground states at the beginning of the introduction.

We now reference what out-performed means in the context of that study (counties where Trump did better in 2016 than Romney did in 2012)

There is an egregious summary statement at the end of the abstract, and repeated twice near the end of the article, but by that time there has been sufficient explanation that there is less chance for the reader to misinterpret. The offending but cogent statement is "Collectively, this data exhibits a strong pattern: counties that voted Republican in the 2016 election are 'sicker' than those that voted Democrat." The authors are referring to the finding that counties where the majority of votes were for the Republican presidential candidate, were also the counties where a higher percentage of residents were in poor physical health, by a number of public health measures, compared to counties where the candidate of the Democrats won the majority vote. (Whether the ailing residents actually voted and who they voted for, we do not know.)

The authors must be aware that another, more pejorative meaning of 'sicker' is "morally unsound or corrupt" (Webster). If this article is accepted, it will be published after a hotly contested national presidential election where weeks later emotions continue to run high and misinterpretations can feed the existing political' and cultural fracture. The summary statement of the authors is a headline=grabbing description of this study's collective findings, but it does not belong in a scientific article because of the possible derogatory interpretation of a cultural and political group. when more accurate and precise wording is possible.

We removed all mention of “sicker” and instead say “worse health outcomes”.

Indeed, the authors do clearly define their variables in the Methods section and appropriately label scatter plots and the supplemental box plot figures in the Analysis section of the article. (Note also that the authors provide a comprehensive list of relevant references and available sources of data used in this research,.) But, by waiting until page 5 in the manuscript to define their terms, the authors allow misunderstanding to take root in the Abstract and introductory sections so that it is possible for someone to quote a passage out of context and undercut belief in an unbiased science. A good copy editor should be able to flag all these instances where the reader could misinterpret the terms and associated results, and improve clarity.

In general, the ample data analysis in this study supports the conclusion that those counties where the Republican presidential candidate won a majority of the popular vote (i.e. Republican counties) are counties that have a greater proportion of residents considered to be ailing or in poor health, and not doing as well economically as residents of those counties where the candidate of the Democrats garnered the majority of votes for President. This study is very complete in showing the correlations between a large number of a health and community well-being variables commonly used in the public health field and the voting patters of the two main political parties. It is the consistency of patterns in those collective findings that makes the study's finding convincing.

To their credit, the research team took two additional steps in analysis that enhance the readers' understanding of the relationship between voting patterns and health status at the county level. The first additional step was to examine the data for discontinuities as revealed by a positive or negative shift in in the majority votes for a political party's presidential candidate, 2016-2012. The second step was to examine trends in a county's well-being and health variables over time. This addresses questions of whether conditions in the county are better now or worse than before, Is there more or less of a particular dynamic now, and if so, how does that trend relate to voting patterns in the presidential election?

The authors do cite some limitations on this research study, but could further strengthen this article by adding a brief discussion about two other possible limitations on the interpretations of their results, specifically, acknowledging the possible role of additional health care access variables other than insurance, and the influence that demographic change in rural areas and small towns of the USA has on the assessment of health status. Both could influence interpretations drawn from the data analysis.

We added to our limitations section to reflect these very big limitations.

In the USA, over time, more and more people 20-39yrs have moved away from small towns and rural areas, leaving older family members, and fewer younger families and children remaining in the area. There has been little replacement, or in-migration to offset the county's loss. A population with this changed age structure has more chronic illness than a population that maintains a more normal age distribution. Such demographic change has been accompanied by decaying downtown business areas, fewer jobs, and less innovation in those same small towns and rural areas, changes which breed deaths of despair. Drug addiction, and suicide are responses to less economic hope and few support structures. Is it possible that the finding that those voting for the Republican presidential candidate are 'sicker', is just a twist on the well-known fact that Trump won the vote of older, conservative, and rural Americans in 2016 and were chiefly responsible for his success in the electoral college vote?

We now stress the important differences between rural/urban settings and how changing demographics affect health.

The last decade has produced many public health studies demonstrating the importance of health care access to the health of an individual and a community. Health care costs and insurance to offset those costs are an important part of access, The study team wisely included a health care insurance variable in their analysis and the authors detail those findings in the article . But, ability to pay is not the sole determinant of health care access. The actual presence of facilities and trained health personnel are necessary to provide not only emergency, acute and chronic medical care for physical and mental health crises, but also rehabilitation for stroke, heart attack, fracture, and trauma -- problems frequently seen in counties with an older, rural, or small own population, and decaying infrastructure. However, in recent news reports, we have learned of the closing of more hospitals that provide services to small towns and rural areas, Not only facilities are few in number, fewer trained health professionals are are located there. Nurses, health educators, primary care providers, as well as those in geriatrics, mental health, pulmonology and other medical specialties are mainly located in or near urban areas with large populations.

We now more extensively discuss the issues of healthcare access as well as healthcare quality, and how these are important variables that need to be studied further, and their partial omission from this study is a limitation.

The authors point out that 'Republican counties' have a higher proportion of residents with behavioral health problems than in 'Democratic counties'. Behavioral health problems can by impacted by access to education. counseling, group support, mentoring practices, and program sponsored rewards and other incentives. Facilities and trained health professionals are needed to promote behavioral health through classes on stopping smoking, weight management, diabetes control and nutrition, stress management, balance and strength raining, grief support, and small group activities designed to foster interchange and reduce social isolation. Yet small towns rand rural areas have limited access to community health or social programs sponsored by health facilities, non-profits, local agencies university or government. Transport itself can be a problem depending on how far away a program is being held. No wonder the emphasis on self-reliance and do-it-yourself. Who and what is there to be of help to most small town and rural folk? The answer is kin (what there is left of them), neighbors and the church.

The question remains: Would the 'Republican counties have a 'sicker' population with more risk factors, (e.g. smoking, obesity, and lack of physical activity), if they had access to the same or similar health care resources as the counties where a Democrat presidential candidate won the the majority vote? Such additional access variables as presence and number of health care facilities, trained health personnel, and transport time to care were outside the focus of this particular research study, but are possible limitation on interpretation of study findings.

We now mention in our limitations section an explanation for why behavioral health could differ between Republican and Democrat counties for many of the reasons presented above (i.e less public health education, group support...)

The final pages of text pages 45-48, begin to explore a fascinating topic--patterns between health policy and political party affiliation, and the current toxic influence of partisanship on health funding, practices, and outcomes, including a short discussion of issues around the 2020 coronavirus pandemic. In effect, the authors start a second, but connected research question and discussion. What the authors have to say is very worthwhile and needs to be pursued as another research article, and draft article detailing question, methods, analysis and conclusions -- or possibly adapt for an Op-Ed in a major national newspaper. I hope the authors will pursue this inquiry. They have the interest, background knowledge, and have done the groundwork. But, the topic and existing associated text should be considered a discrete inquiry and discussion, separate from this article on the relationship of a county population's health and social well-being and their voting patterns in 2012 and 2016 presidential election.

We have removed the sections of text referred to on pages 45-48 above.

---

## [Decision Letter · Decision Letter 1]

19 Apr 2021

PONE-D-20-17485R1

Viewing the US presidential electoral map through the lens of public health.

PLOS ONE

Dear Dr. Hamamsy,

Thank you for submitting your manuscript to PLOS ONE. After careful consideration, we feel that it has merit but does not fully meet PLOS ONE’s publication criteria as it currently stands. Therefore, we invite you to submit a revised version of the manuscript that addresses the points raised during the review process.

Please address the concerns and recommendations of Reviewer 2. Specifically  (1) the main question, the whole rationale of this paper is too vague. What do the authors hope to accomplish? What difference might the findings make in real life?  In the Abstract, the authors say “it is important to understand the relationship between voting “patterns, health, disease, and mortality.”  Why? How do you measure “understand”?  (2) How much of the associations between voting patterns, health, disease and mortality can be explained by social economic and demographic factors? (3) Since voting patterns are associated with health behaviors and health outcomes, the results of this study might indicate, for example, need for funding of special health education initiatives for rural, older, economically marginal citizens.

We look forward to receiving your revised manuscript.

Kind regards,

M. Harvey Brenner, PhD

Academic Editor

PLOS ONE

Reviewers' comments:

Reviewer's Responses to Questions

**Comments to the Author**

1. If the authors have adequately addressed your comments raised in a previous round of review and you feel that this manuscript is now acceptable for publication, you may indicate that here to bypass the “Comments to the Author” section, enter your conflict of interest statement in the “Confidential to Editor” section, and submit your "Accept" recommendation.

Reviewer #1: All comments have been addressed

Reviewer #2: (No Response)

2. Is the manuscript technically sound, and do the data support the conclusions?

Reviewer #1: Yes

Reviewer #2: Partly

3. Has the statistical analysis been performed appropriately and rigorously? 

Reviewer #1: Yes

Reviewer #2: Yes

4. Have the authors made all data underlying the findings in their manuscript fully available?

Reviewer #1: Yes

Reviewer #2: Yes

5. Is the manuscript presented in an intelligible fashion and written in standard English?

Reviewer #1: Yes

Reviewer #2: Yes

6. Review Comments to the Author

Reviewer #1: I remain concerned about too much data. Table 1 is 40 pages long. Maybe much of this should be placed in an appendix. But that is an editor call. I also wonder how much variation the first two principle components accounted for. And what is your interpretation of each of these two? It would be easy to add this.

Finally, I want to emphasize the importance of this work. The polarization in our society is the worst I have seen since the Viet Nam era. The Democrats lost much of the white working class vote and Trump hijacked the Republican party with it. More understanding of their plight seems a national priority. This paper serves that end.

Reviewer #2: I recognize the completeness of the revisions that the authors have made. However, I have more concerns, and they follow in my Comments to the Authors, after first acknowledging the adequate sections of the revised paper. My comments total 26,861 characters or 22,671 excluding spaces, so I will upload my review comments as an attachment.

7. PLOS authors have the option to publish the peer review history of their article (what does this mean?). If published, this will include your full peer review and any attached files.

Reviewer #1: **Yes: **Peter Hilsenrath

Reviewer #2: **Yes: **Dory Storms, ScD

---

## [Author Response · Author response to Decision Letter 1]

12 May 2021

We would like to thank the editor for closely reading this manuscript as well as the reviewer comments and our responses to them. In addition to addressing reviewer comments we have streamlined the overall text and links to the supplemental material. We detail our full response to the review comments below.

Reviewer 1:

We want to thank the reviewer for all of their helpful, productive comments in this process.

We have addressed reviewer #1’s point to include the variance explained of the first 2 principle components, as well as noting the variables that are associated with those principle components. We have also removed the supplementary tables from the manuscript, and into the appendix.

Reviewer 2:

We want to thank the reviewer for their expertise in guiding the process of refining this paper and we appreciate the reviewer’s kind words.

We appreciate their comment regarding the clarity of the paper and have taken steps to improve the general understanding of the goal of our work.

We agree that the majority of tables should be included as an appendix and have modified the manuscript so they are not in the body of the manuscript. 

We particularly appreciate the reviewer’s specific comments regarding how to make this work applicable beyond the realm of pure academic interest. We are similarly interested in how this sort of work can have actionable consequences. As such, we have modified the manuscript in particular sections to reflect our belief that these data should guide policy.

---

## [Decision Letter · Decision Letter 2]

18 Jun 2021

Viewing the US presidential electoral map through the lens of public health.

PONE-D-20-17485R2

Dear Dr. Tymor Hamamsy,

We’re pleased to inform you that your manuscript has been judged scientifically suitable for publication and will be formally accepted for publication once it meets all outstanding technical requirements.

Kind regards,

M. Harvey Brenner, PhD

Academic Editor

PLOS ONE

Additional Editor Comments (optional):

Reviewers' comments:

Reviewer's Responses to Questions

**Comments to the Author**

1. If the authors have adequately addressed your comments raised in a previous round of review and you feel that this manuscript is now acceptable for publication, you may indicate that here to bypass the “Comments to the Author” section, enter your conflict of interest statement in the “Confidential to Editor” section, and submit your "Accept" recommendation.

Reviewer #2: All comments have been addressed

2. Is the manuscript technically sound, and do the data support the conclusions?

Reviewer #2: Yes

3. Has the statistical analysis been performed appropriately and rigorously? 

Reviewer #2: Yes

4. Have the authors made all data underlying the findings in their manuscript fully available?

Reviewer #2: Yes

5. Is the manuscript presented in an intelligible fashion and written in standard English?

Reviewer #2: Yes

6. Review Comments to the Author

Reviewer #2: PAGE 25, SECOND PARAGRAPH NEEDS CITATION NUMBER. JUST SAYS (CITATION)!

This is a beautiful paper! Clear, careful wording . Your hard work at revising has resulted in a first-class paper with implications for state, county, and national level politicians of both parties for how best to strengthen county public health policy and funding.

7. PLOS authors have the option to publish the peer review history of their article (what does this mean?). If published, this will include your full peer review and any attached files.

Reviewer #2: No

---

## [Editor Report · Acceptance letter]

30 Jun 2021

PONE-D-20-17485R2 

Viewing the US presidential electoral map through the lens of public health. 

Dear Dr. Hamamsy:

I'm pleased to inform you that your manuscript has been deemed suitable for publication in PLOS ONE. Congratulations! Your manuscript is now with our production department. 

Kind regards, 

on behalf of

Professor M. Harvey Brenner 

Academic Editor

PLOS ONE